# Topology-aware Graph Diffusion Model with Persistent Homology

**Joonhyuk Park**[1*], **Donghyun Lee**[2*], **Yujee Song**[2], **Guorong Wu**[3], **Won Hwa Kim**[1]
[1]POSTECH    [2]Samsung Eletronics    [3]UNC-Chapel Hill
{pjh1023, wonhwa}@postech.ac.kr

## Abstract

Generating realistic graphs faces challenges in estimating accurate distribution of graphs in an embedding space while preserving structural characteristics. However, existing graph generation methods primarily focus on approximating the joint distribution of nodes and edges, often overlooking topological properties such as connected components and loops, hindering accurate representation of global structures. To address this issue, we propose a Topology-Aware diffusion-based Graph Generation (TAGG), which aims to sample synthetic graphs that closely resemble the structural characteristics of the original graph based on persistent homology. Specifically, we suggest two core components: 1) Persistence Diagram Matching (PDM) loss which ensures high topological fidelity of generated graphs, and 2) Topology-aware Attention Module (TAM) which induces the denoising network to capture the homological characteristics of the original graphs. Extensive experiments on conventional graph benchmarks demonstrate the effectiveness of our approach indicating high generation performance across various metrics, while achieving closer alignment with the distribution of topological features observed in the original graphs. Furthermore, application to real brain network data showcases its potential for complex and real graph applications.

## 1 Introduction

The major goal of graph generation is to achieve high resemblance between generated graphs and their reference counterparts. To achieve this goal, various graph generation approaches have been studied based on conventional generative models, e.g., recurrent neural networks [47], variational autoencoders [38] and diffusion models [23, 43], and each exhibited promising results. Despite the achievement in the context of the quantitative measures, e.g., similarity in distributions of graph characteristics such as degree and clustering coefficients, there remains a limitation on generating graphs coherent to the graph structure via the lens of graph topology.

Brain network is perhaps a suitable example to demonstrate the challenges above. A brain network characterizes intricate wiring system of the brain, which is represented as a graph with anatomical regions of interest (ROIs) defining its nodes and the connectivity between different ROIs serving as edges [7, 16]. It is often large and dense, and its topological properties are well-known critical biomarkers [35, 39]. Moreover, brain networks are expensive; acquiring diffusion magnetic resonance images (dMRI) and processing them via tractography [42] to obtain structural brain connectome is costly in both cost and labor. In this regime, generating realistic graphs (e.g., brain networks) that preserve their inherent connectivity as well as global structures is highly demanding.

In recent years, various graph generative models have been heavily studied, but they often fall short in capturing essential topological features crucial for modeling interconnected brain regions with high

---

*J. Park and D. Lee contributed equally. Work done while D. Lee[2] and Y. Song[2] were at POSTECH.

39th Conference on Neural Information Processing Systems (NeurIPS 2025).

fidelity. The methods in [23] and [32] proposed score-based diffusion methods in a continuous time domain, originally defined for images [40]. However, the continuous diffusion methods suffer from high computational cost, as the forward and reverse diffusion process is performed on infinitesimal continuous time point. Moreover, the uniformly added Gaussian noise results in a noisy and complete graph, which causes the loss of structural information, e.g., sparsity of a graph. Later, [43] proposed a discrete diffusion method, applying additive noise to each node and edge independently for graphs, nevertheless, existing methods overlook the *topologically invariant characteristics*, e.g., geometric shape and connectivity, limiting the generation.

To overcome such issues, we propose a novel **T**opology-**A**ware **G**raph **G**eneration (TAGG), in which the sampled graphs resemble not only in the distributions of the original graphs in the embedding space but also in the *homological features* of the original graphs. Conventionally, topological data analysis (TDA) from algebraic topology has been studied in various graph analyses [6, 9, 21, 44] to investigate topological features, and we bridge the gap between TDA and graph diffusion model to generate topologically realistic graphs. We define Persistence Diagram Matching (PDM) loss with persistence homology, which regularizes homological features of the reference graphs to be incorporated in graph generation process via 1-Wasserstein distance. Furthermore, we introduce a Topology-aware Attention Module (TAM) which utilizes persistence landscape [5] of a given graph to foster the denoising network with global structural information.

**Contributions.** To this end, our main contributions are summarized as follows: **1)** We propose a novel topology-aware graph generation method that yields homologically similar graphs with high structural fidelity. **2)** We propose PDM loss, utilizing persistent homology to encode the graph topology. **3)** We propose a Topology-aware Attention Module (TAM) that leverages persistence landscape to enhance the denoising network in capturing graph topology.

Our model demonstrates superior performance on real and synthetic graph generation, with intuitive visualizations for topological comparisons. Especially with the application on brain network generation from Alzheimer's Disease Neuroimaging Initiative (ADNI), our method demonstrates its adaptability to diverse real-world graph generation tasks.

## 2 Related work

**Graph Generation.** Graph generation has been developed in two major branches; autoregressive and one-shot. Auto-regressive methods [4, 22, 25, 38, 46, 47] recursively capture the intricate graph dependencies, and sequentially generate the graph structure conditioned on the current incomplete graph. In spite of their impressive performance, auto-regressive approaches exhibit considerable computational demands due to the increasing number of generation steps along with the graph size. Also, they face a challenge stemming from the absence of an inherent node generation order. Conversely, one-shot methods [12, 27, 28, 48] generate the whole graph, i.e., every node and edge, at once. By doing so, they reduce computational requirements while facing performance degradation as the dataset scale grows. Recently, diffusion-based methods [4, 11, 23, 29, 30, 43] showed promising capability in graph generation, by defining the forward and reverse diffusion processes and training a neural network that mimics the reverse process to reconstruct the graphs.

**Persistent Homology.** Persistent homology from computational topology studies the topological features of given objects, such as the number of holes [15]. It provides a way to capture and quantify the shapes and global structures by computing homological features of objects. By treating a graph as a topological object, the concept of persistent homology can be utilized to analyze the global structure of graphs, which leads improvements for graph classification [18, 20, 49] and link prediction [45]. Furthermore, persistent homology has been successfully applied in biology [6, 10, 33], signal processing [44], and point cloud [31], demonstrating its versatility and robustness.

## 3 Preliminaries

We briefly review persistent homology, which extracts homological properties from objects. We refer readers to [8] and [15], should further questions arise regarding topological data analysis.

**Simplicial Complex.** Let $V$ be a non-empty set. A simplicial complex $K$ is a collection of non-empty finite subsets of $V$ which satisfies the following two properties; (1) for any $v \in V$, $\{v\} \in K$, and (2) if $\sigma \in K$ and $\tau \subseteq \sigma$, then $\tau \in K$. An element of $K$ is called a *simplex* and the dimension of a simplex is determined by the length of its elements. For example, an element $\tau \in K$ with $|\tau| = k + 1$

is a $k$-simplex whose dimension is $k$. The dimension of a simplicial complex $K$ is defined by the highest-dimension of its simplices.

**Graph as a Simplicial Complex.** Consider an undirected graph $G = (V, E)$, where $V$ is a set of $N$ nodes and $E \subseteq V \times V$ is a set of edges. Then, a graph $G$ can be interpreted as a 1-dimensional simplicial complex $K_G$ whose 0-simplices are the nodes and 1-simplices are the edges, i.e.,

$$G = K_G = \{\{v\} : v \in V\} \cup E. \tag{1}$$

The homological properties of a graph $G$ can be characterized by its connected components and loops, each corresponding to the 0- and 1-dimensional features, respectively. These properties are fundamental in capturing the underlying topological structure of the graph. The quantity of such properties in a given dimension $k$ is referred to as the *Betti number* $\beta_k$, which serves as a key descriptor of the topological characteristics of the graph.

**Filtration.** Filtration of a graph $G$ is a sequence of nested subgraphs of $G$, i.e., $\emptyset = G^{(0)} \subseteq G^{(1)} \subseteq G^{(2)} \subseteq \ldots \subseteq G^{(N-1)} \subseteq G^{(N)} = G$. Specifically, the filtration of $G$ can also be defined using a 0-simplex (i.e., vertex) filter function $f : V \to [0, \infty)$ on node degree $\deg(\cdot)$, as in [19]:

$$\forall \{v\} \in G, \quad f(\{v\}) := \deg(\{v\}) / \max_{\{v'\} \in G} (\deg(\{v'\})). \tag{2}$$

Suppose that the computed filter values $a_i$ are given in an ascending order $0 = a_0 < a_1 < a_2 < \cdots < a_N$, where $a_i \in \{f(\{v\}) : \{v\} \in G\}$. To define the filtration of a graph $G$, we adopt a non-negative scale parameter $\epsilon$, which is incrementally increased from 0. Upon reaching $\epsilon = a_1$, we construct $G^{f,1}$ from $G^{f,0} = \emptyset$ by adding the node $v_1$. When $\epsilon$ subsequently reaches at $a_2$, we extend $G^{f,1}$ to $G^{f,2}$ by adding the node $v_2$ and the edge connecting $v_2$ and the nodes in previous subgraph, i.e., $G^{f,1}$, if the edge exists. By repeating this process, we systemically define the sublevel set filtration induced by $f$ for $0 \leq i \leq N$ as:

$$G^{f,i} = \{\sigma \in G : \max_{v \in \sigma} f(v) \leq a_i\} = f^{-1}([0, a_i])$$
$$\emptyset = G^{f,0} \subseteq G^{f,1} \subseteq G^{f,2} \subseteq \cdots \subseteq G^{f,N-1} \subseteq G^{f,N} = G. \tag{3}$$

**Persistent Homology.** The homological features can be extended via tracking the filtration of $G$. Filtration leads to the notion of persistent homology by monitoring the (de)formation of homological features in each $G^{f,i}$ along the filtration, thereby allowing us to obtain their homological relevance in a given dimension, i.e., how long each homological feature persists. Specifically, if a homological feature, e.g., a connected component, first appears at $G^{f,i}$ and disappears at $G^{f,j}$ in 0-dimension, its *birth* and *death* are defined as $i$ and $j$, respectively.

**Representation of Persistent Homology.** Suppose that a homological feature is born at $G^{f,i}$ and dies at $G^{f,j}$, i.e., that it persists from $i$ to $j$. It can be denoted as a tuple of birth and death pair, i.e., $(i, j)$, known as *persistence barcode*. By considering each $i$ and $j$ as coordinates and plotting the barcodes $(i, j)$ on the $\mathbb{R}^2$ plane, we can obtain a *persistence diagram* $\mathcal{D}_G$ of the graph $G$:

$$\mathcal{D}_G = \{(b, d) : (b, d) \text{ is a barcode of } G\} \subseteq \mathbb{R}^2. \tag{4}$$

Note that a persistence diagram $\mathcal{D}_G$ can be obtained separately with respect to the dimension of the barcodes, i.e., the dimension of the homological feature a barcode encodes. For every $n$ points $(b, d) \in \mathcal{D}_G$, i.e., $|\mathcal{D}_G| = n$, we define a piece-wise linear function $f_{(b,d)} : \mathbb{R} \to [0, \infty)$, as:

$$f_{(b,d)}(x) = \begin{cases} 0 & \text{if} \quad x \notin (b, d), \\ x - b & \text{if} \quad x \in \left(b, \frac{b+d}{2}\right], \\ d - x & \text{if} \quad x \in \left(\frac{b+d}{2}, d\right). \end{cases} \tag{5}$$

*Persistence landscape* [5] of a persistence diagram $\mathcal{D}_G$ can be established as a sequence of functions $\lambda_l : \mathbb{R} \to [0, \infty)$ for $l \in \mathbb{N}$, where $\lambda_l(x)$ denotes the $l^{\text{th}}$ largest value of the functions $f_{(b_i, d_i)}(x)$, for $i \leq n$. We obtain $s$ points by dividing the domain of the function $\lambda_l(x)$ where $\lambda_l(x) \geq 0$ into $(s + 1)$ equal sub-intervals. With chosen $L$ and $S \in \mathbb{N}$, we can obtain a homological feature vector $\mu_G \in \mathbb{R}^{L \cdot S}$ for a given graph $G$ as follows:

$$\mu_G^l = [\lambda_l(x_1), \ldots, \lambda_l(x_s), \ldots, \lambda_l(x_S)] \in \mathbb{R}^S$$
$$\mu_G = [\mu_G^1, \ldots, \mu_G^l, \ldots, \mu_G^L] \in \mathbb{R}^{L \cdot S}, \tag{6}$$

where $1 \leq l \leq L$, and $1 \leq s \leq S$. The persistence diagram $\mathcal{D}_G$ and the homological feature vector $\mu_G$ encode the entire information about persistent homology of a given graph [5, 15].

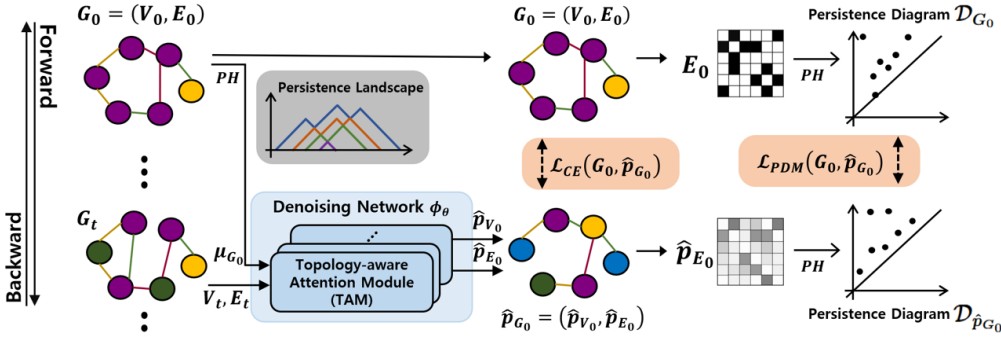

Figure 1: Training of the denoising network $\phi_\theta(G_t, \mu_{G_0})$. This network takes a noisy graph $G_t$ and the embedding $\mu_{G_0}$ obtained from persistence landscape of the original graph $G_0$ as an input, and outputs the probability vector of nodes and edges, $\hat{p}_{V_0}$ and $\hat{p}_{E_0}$, to predict $G_0$. During training, we utilize these predictions in two ways: **1)** cross-entropy loss $\mathcal{L}_{CE}$ over all nodes and edges, and **2)** Persistence Diagram Matching loss $\mathcal{L}_{PDM}$ which computes the discrepancy between persistence diagrams of $G_0$ and $\hat{p}_{G_0}$.

## 4  TAGG: Topology-aware graph generation

Unlike conventional diffusion-based graph generation methods which utilize Gaussian noise in continuous space [23, 32], we perform the diffusion process in discrete space to preserve the sparsity of the graph for every diffusion time steps. We follow the settings in [43], which successfully expanded the method in [3] for generating graphs with categorical node and edge attributes by treating each node and edge as a categorical random variable.

Let $G_0 = (V_0, E_0)$ be the original graph with $N$ nodes, where $V_0 \in \mathbb{R}^{N \times F_V}$ and $E_0 \in \mathbb{R}^{N \times N \times F_E}$ are the node and edge matrices with $F_V$ and $F_E$ attributes, respectively. At time step $t$, we denote the attribute of the $i$-th node $v^i$ as a one-hot vector $v_t^i \in \mathbb{R}^{F_V}$ and that of the edge $e^{i,j}$ between $v^i$ and $v^j$ as a one-hot vector $e_t^{i,j} \in \mathbb{R}^{F_E}$. Thus, each element of $V_t$ and $E_t$ is represented as a one-hot vector.

### 4.1  Forward process

Considering node and edge attributes as one-hot vectors, we follow the settings of [3] and [43] to define the forward and reverse process of diffusion acting on the node and edge attributes. We denote the forward diffusion process of each time step to impose noise as transition matrices $Q_t$, where $t = 1, 2, \ldots, T$, and each element of the matrices, $[Q_t]_{\eta, \xi}$, represents the probability that state $\eta$ changes to state $\xi$ as the time step changes from $t-1$ to $t$, i.e., $\left[Q_t^V\right]_{\eta^V, \xi^V} = q(v^t = \xi^V \mid v^{t-1} = \eta^V)$ and $\left[Q_t^E\right]_{\eta^E, \xi^E} = q(e^t = \xi^E \mid e^{t-1} = \eta^E)$.

From time step $t - 1$, a noised graph $G_t$ can be obtained by sampling the type of nodes and edges from the categorical distribution after transition, which is derived as:

$$q\left(G_t \mid G_{t-1}\right) = \left(V_{t-1} Q_t^V, E_{t-1} Q_t^E\right). \tag{7}$$

Specifically, the transition matrix $Q_t^V$ is determined by the dimension of node categories, i.e., $Q_t^V = (1 - \beta_t)\boldsymbol{I} + (\beta_t / F_V)\boldsymbol{J}_{F_V}$, where $\boldsymbol{I} \in \mathbb{R}^{F_V \times F_V}$ is an identity matrix and $\boldsymbol{J}_{F_V} \in \mathbb{R}^{F_V \times F_V}$ is a matrix of ones, and $\beta_t$ is a real value in range $[0, 1]$. The transition matrix for edge $Q_t^E$ is determined in the same manner.

Assuming Markovian property of the process, we can derive the transition matrix from time 0 to time $t$ by simply multiplying each transition matrices: $\bar{Q}_t^V = Q_1^V \cdot Q_2^V \cdots Q_t^V$, and $\bar{Q}_t^E = Q_1^E \cdot Q_2^E \cdots Q_t^E$. Then, similar to Eq. (7), the noised graph $G_t$ can also be obtained from time 0 by sampling from $q\left(G_t \mid G_0\right) = \left(V_0 \bar{Q}_t^V, E_0 \bar{Q}_t^E\right)$. Details on the forward process are provided in Appendix A.1.

### 4.2  Topology-aware denoising network

In this section, we introduce a topology-aware graph denoising network $\phi_\theta$ parametrized by $\theta$, which estimates the probability vector $\hat{p}$ of the nodes and edges of the original graph $G_0$, i.e., $\hat{p}_{G_0} = (\hat{p}_{V_0}, \hat{p}_{E_0})$. In addition to the noisy graph $G_t$, we leverage the homological feature $\mu_{G_0}$ of $G_0$ obtained via Eq. (6) as an input of $\phi_\theta$ to retain the topological structure of the original graph during estimation.

For this, we introduce a novel Topology-aware Attention Module (TAM), which serves as a fundamental component within the denoising network $\phi_\theta$. Given a noisy graph $G_t$ and $\mu_{G_0}$, TAM is iteratively applied to refine the node and edge attributes within $\phi_\theta$. TAM operates as illustrated in Fig. 2: at the $m$-th iteration, node and edge attributes $V_t^m$ and $E_t^m$ are inputted to TAM, which produces refined output $V_t^{m+1}$ and $E_t^{m+1}$ as $(V_t^{m+1}, E_t^{m+1}) = \text{TAM}(V_t^m, E_t^m, \mu_{G_0})$. Ultimately, the denoising network $\phi_\theta$ estimates the

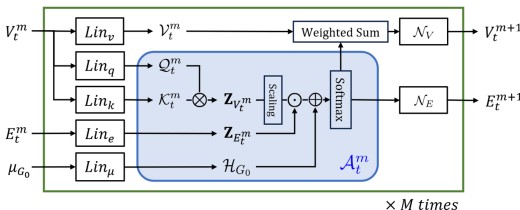

Figure 2: Topology-aware Attention Module (TAM). $\otimes$ and $\odot$ denote the outer and element-wise product. The blue rectangle is the topology-aware attention score $\mathcal{A}_t^m$.

categorical probability vector of the original graph, i.e., $\hat{p}_{G_0} = (\hat{p}_{V_0}, \hat{p}_{E_0})$, by iteratively applying TAM $M$ times as:

$$(\hat{p}_{V_0}, \hat{p}_{E_0}) = \phi_\theta(G_t, \mu_{G_0}) = \text{TAM}^M(V_t^0, E_t^0, \mu_{G_0}), \tag{8}$$

where $m = 0, 1, \cdots, M-1$, and the initial conditions are given by $V_t^0 = V_t$ and $E_t^0 = E_t$.

In TAM, the node attribute $V_t^m$ is passed through three distinct linear transformations to produce the query $\mathcal{Q}_t^m$, key $\mathcal{K}_t^m$, and value $\mathcal{V}_t^m$ matrices. Similarly, the edge embedding $\mathbf{Z}_{E_t^m}$ and the homological embedding $\mathcal{H}_{G_0}$ are obtained by applying a linear transformation to $E_t^m$ and $\mu_{G_0}$, respectively. After obtaining these representations, the topology-aware attention score $\mathcal{A}_t^m$ is computed as:

$$\mathcal{A}_t^m = \text{softmax}\left( \frac{\mathcal{Q}_t^m \cdot \mathcal{K}_t^{mT}}{\sqrt{d_k}} \cdot \mathbf{Z}_{E_t^m} + \mathcal{H}_{G_0} \right), \tag{9}$$

where $d_k$ is the dimension of the query and key matrices.

Using the topology-aware attention score $\mathcal{A}_t^m$, we obtain the updated node and edge attributes, i.e., $V_t^{m+1}$ and $E_t^{m+1}$, with additional linear transformations $\mathcal{N}_V$ and $\mathcal{N}_E$ as:

$$V_t^{m+1} = \mathcal{N}_V\left(\mathcal{A}_t^m \mathcal{V}_t\right), \quad E_t^{m+1} = \mathcal{N}_E\left(\mathcal{A}_t^m\right). \tag{10}$$

Notice that TAM further adds the homological embedding $\mathcal{H}_{G_0}$ as a global attention bias term to induce the attention to capture the graph's topological features. Conventional graph transformers multiply the edge embeddings to the node-wise attention computation, i.e., multiplication of the query and key matrices, thereby leveraging edge information [14]. However, as edge information represent local structures, utilizing the edge embedding alone may be insufficient for incorporating global structural information into the attention score. To address this limitation, TAM integrates the homological embedding $\mathcal{H}_{G_0}$ as a bias term to inject global structural information into the attention.

Since $\mu_{G_0}$ encodes the homological information by tracking every subgraph in the filtration of $G_0$ (from Eq. (3)), it contains rich structural information that is challenging for the network to capture solely from the final subgraph of the filtration, i.e., the original graph. Therefore, incorporating $\mathcal{H}_{G_0}$, the homological embedding obtained from $\mu_{G_0}$, as a global attention bias term enables TAM to estimate topology-aware node and edge embeddings, facilitating the final estimation of the topology-aware probability vector $\hat{p}_{G_0}$. Consequently, the integration of $\mu_{G_0}$ bolsters the denoising network $\phi_\theta$ in generating more realistic graphs.

Moreover, it is noteworthy that the homological feature vector $\mu_{G_0}$ can be pre-computed during the data preprocessing step, thereby minimizing the need for additional computation for training. We empirically demonstrate in Sec. 5 that the homological feature vector $\mu_{G_0}$ helps the model to better learn the original distribution of nodes and edges.

### 4.3 Training objective of TAGG

To produce accurate estimation of the probability vector $\hat{p}_{G_0}$, we optimize the denoising network $\phi_\theta$ with two loss terms: 1) Persistence Diagram Matching loss, which aligns the homological features of the generated graphs, and 2) Cross Entropy (CE), which ensures the node and edge attributes of the generated graphs to resemble those of original graphs.

Let us first introduce **Persistence Diagram Matching loss** $\mathcal{L}_{PDM}$. As discussed in Sec. 3, persistence diagrams hold comprehensive homological information of graphs obtained via persistent homology.

In order to ensure the generated graphs to resemble the homological features of the original graphs, which is our main contribution, we aim to minimize the discrepancy between the persistence diagrams of the original and the generated graph.

Given the original graph $G_0$ and the estimate $\hat{p}_{G_0}$, the persistence diagrams $\mathcal{D}_{G_0}$ and $\mathcal{D}_{\hat{p}_{G_0}}$ can be computed, as defined in Eq. (4). Considering the diagrams as distributions [26], we compute the discrepancy between the two distributions via 1-Wasserstein distance $W_1(\cdot)$ as:

$$\mathcal{L}_{PDM}(G_0, \hat{p}_{G_0}) = W_1(\mathcal{D}_{G_0}, \mathcal{D}_{\hat{p}_{G_0}}) = \inf_{\pi}\left(\sum_{\boldsymbol{x} \in \mathcal{D}_{G_0}} ||\boldsymbol{x} - \pi(\boldsymbol{x})||\right), \tag{11}$$

for any bijection $\pi : \mathcal{D}_{G_0} \to \mathcal{D}_{\hat{p}_{G_0}}$.

Note that the bijection $\pi$ between two persistence diagrams holds the following two challenges: 1) the bijection does not exist if the persistence diagrams have different number of points, which is true in most cases, and 2) the matching between points from two different persistence diagrams may be misleading, i.e., the two matched points may hold homological features of different dimension, such as matching a connected component to a loop. Hence, following the common approach in TDA [24], we pad the diagrams with points on the diagonal to ensure a proper matching between points of the two persistence diagrams. Also, as described in Sec. 3, persistence diagrams for each dimension can be acquired separately by plotting the barcodes of each dimension. Therefore, the PDM loss is computed over $\mathcal{D}_{G_0}$ and $\mathcal{D}_{\hat{p}_{G_0}}$ of the same dimension, with the bijection guaranteed to match points with homological features of the same dimension. The final $\mathcal{L}_{PDM}$ is determined by summing the distances across all dimensions.

In addition, we guide the probability vectors from $\phi_\theta$ to approximate the ground truth attributes of $G_0$. This is conventionally done by minimizing the Cross Entropy $CE(\cdot)$ loss over all nodes and edges:

$$\begin{aligned}\mathcal{L}_{CE}(G_0, \hat{p}_{G_0}) &= \mathcal{L}_{CE}^V(G_0, \hat{p}_{G_0}) + \alpha_1 \mathcal{L}_{CE}^E(G_0, \hat{p}_{G_0}) \\ &= \sum_{1 \le i \le N} CE\left(v^i, \hat{p}_{v_t^i}\right) + \alpha_1 \sum_{1 \le i,j \le N} CE\left(e^{i,j}, \hat{p}_{e_t^{i,j}}\right),\end{aligned} \tag{12}$$

where $\alpha_1 \in (0, 1]$ is a real value. The final training objective linearly combines the two losses: $\mathcal{L}_{final} = \mathcal{L}_{CE}(G_0, \hat{p}_{G_0}) + \alpha_2 \mathcal{L}_{PDM}(G_0, \hat{p}_{G_0})$, for a real value $\alpha_2 \in (0, 1]$. During the training process, $\mathcal{L}_{PDM}$ helps the network to learn the topological structure of the original graphs. The overall training framework is shown in Fig. 1 and the effect of $\mathcal{L}_{PDM}$ is validated in Sec. 5. For further details of TAGG, refer to Appendix A.2.

## 4.4 Reverse process

After optimizing the denoising network $\phi_\theta$, we utilize the reverse process in [43] to generate new graphs. By iteratively estimating the denoised graph $\hat{p}_{G_0}$ from a noisy graph $G_t$ and imposing noise to the estimated graphs $\hat{p}_{G_0}$ by $q(G_{t-1} \mid G_0)$ from $t = T$ to 1, we can synthesize a new graph. Note that unlike in the training step, where each homological feature vector $\mu_{G_0}$ is derived from its corresponding original graph $G_0$, the matching of $\mu_{G_0}$ to its original graph $G_0$ cannot be defined in the reverse process, thus requiring homological feature vectors based on graphs from the training dataset. Hence, we utilize the averaged homological feature vector $\mu_{G'} \in \mathbb{R}^{L \cdot S}$, where $L$ and $S$ are the number of piece-wise linear functions and the number of sub-intervals as introduced in Eq. (6). The average operation is performed over all training graphs.

## 5 Experiments

### 5.1 Dataset and experimental settings

**Graph Benchmark.** To obtain a coherent analysis of graph generation performance, we adopt three conventional benchmarks of real and synthetic graphs: (1) Community-small: 200 synthetic graphs with $12 \le |V| \le 20$ generated from a stochastic block model with two communities, (2) Ego-small: 200 small sub-graphs of the Citeseer network dataset [37] with $4 \le |V| \le 18$, and (3) ENZYMES: 600 protein tertiary structures of the enzymes in graphs from the BRENDA database [36]. Additional experiments on other datasets, i.e., SBM and Planar graphs, are included in the Appendix A.5.

Table 1: Quantitative comparison with baseline models on synthetic and real graph datasets. The best and second best results are highlighted in **bold** and underline, with lower values indicating better performance.

| Method | ADNI Real, $\lvert V \rvert = 160$ | | | | ENZYMES Real, $10 \leq \lvert V \rvert \leq 125$ | | | | Community-small Synthetic, $12 \leq \lvert V \rvert \leq 20$ | | | | Ego-small Real, $4 \leq \lvert V \rvert \leq 18$ | | | |
|---|---|---|---|---|---|---|---|---|---|---|---|---|---|---|---|---|
| | Deg.↓ | Clus.↓ | Orbit↓ | Avg.↓ | Deg.↓ | Clus.↓ | Orbit↓ | Avg.↓ | Deg.↓ | Clus.↓ | Orbit↓ | Avg.↓ | Deg.↓ | Clus.↓ | Orbit↓ | Avg.↓ |
| GraphRNN [47] | 1.392 | 0.916 | **0.153** | 0.820 | 0.161 | 0.942 | 0.112 | 0.405 | 0.183 | 0.182 | 0.113 | 0.159 | 0.069 | 0.090 | 0.052 | 0.071 |
| EDP-GNN [32] | 1.063 | 1.430 | 0.626 | 1.039 | 0.052 | 0.895 | 0.474 | 0.474 | 0.056 | **0.038** | 0.069 | 0.054 | 0.029 | 0.046 | 0.008 | 0.028 |
| GDSS [23] | 0.949 | 1.104 | 0.165 | 0.739 | 0.314 | 0.506 | 0.084 | 0.301 | **0.033** | 0.112 | 0.009 | 0.051 | 0.045 | 0.076 | 0.008 | 0.043 |
| DiGress [43] | 0.504 | 1.168 | 0.379 | 0.683 | 0.023 | 0.051 | 0.205 | 0.093 | 0.089 | 0.091 | 0.049 | 0.076 | 0.026 | 0.090 | 0.023 | 0.046 |
| LocalPPGN [4] | 0.777 | **0.149** | 0.954 | 0.627 | 0.037 | 0.068 | 0.048 | 0.051 | 0.034 | 0.218 | 0.018 | 0.090 | 0.014 | 0.091 | **0.006** | 0.037 |
| TAGG | **0.160** | 0.886 | 0.237 | **0.427** | **0.003** | **0.045** | **0.015** | **0.021** | 0.048 | 0.068 | **0.004** | **0.040** | **0.012** | **0.024** | 0.010 | **0.015** |

**Brain Network.** To validate practicability of TAGG, we use brain connectivity from Alzheimer's Disease Neuroimaging Initiative (ADNI). In house tractography pipeline was applied to Diffusion Weighted Imaging (DWI) of healthy subjects from ADNI adhering to the Destrieux atlas [13] with 160 regions of interest (ROIs) comprising 148 cortical and 12 sub-cortical regions. The dataset is composed of $844$ undirected weighted graphs, i.e., structural brain connectivity, where the edge weights represent the number of fiber tracts connecting different ROIs. The edges were thresholded by removing those below 5%p of the maximum edge weights to obtain sparsity.

**Baselines.** We used the following one-shot deep generative methods as baselines: EDP-GNN [32], GDSS [23], DiGress [43], and the one-shot version of LocalPPGN [4]. In addition, a conventional auto-regressive generation method, i.e., GraphRNN [47], is used for comparison.

**Evaluation Setting.** The (dis)similarity between distributions of graph statistics on the same number of generated and test graphs were computed using the maximum mean discrepancy (MMD) [17]. Specifically, we compared the distributions of degree, clustering coefficient, and the number of occurrences of orbits with 4 nodes, as in [23] and [32]. For consistency, we adhered to the train/test split reference from [23] and performed three replicate experiments to report averaged performance.

## 5.2 Quantitative analysis

**Results.** The comparison between the baselines and the proposed method is shown in Table 1. Note that although the models for the best and the second best performances vary in each metric, TAGG steadily shows highly promising performance, especially in the averaged value of the three MMD metrics. Specifically, we observed that DiGress, the referenced discrete diffusion based graph generation method, did not perform well on small-scale graphs, showing $0.076$ (4th) and $0.046$ (5th) averaged MMD on Community-small and Ego-small dataset, whereas TAGG showed superior performance across all metrics, including the averaged MMD score of $0.040$ (1st) and $0.015$ (1st).

**Ablation study on $\mu_{G_0}$.** We conducted an ablation study to evaluate the impact of the homological feature $\mu_{G_0}$. In Table 2, we show that the feature $\mu_{G_0}$ introduced in Sec. 4.2, enhances the denoising network to generate quantitatively improved realistic graphs by providing the underlying global structural information. Note that utilizing $\mu_{G_0}$ provides performance gain (i.e., from 0.061 to 0.096 decrease in averaged MMD gains) in most categories in Table 2.

**Ablation study on $\mathcal{L}_{PDM}$.** To evaluate the effectiveness of our PDM loss $\mathcal{L}_{PDM}$, we also provide an ablation study in Tab. 3. Similar to the method explained in Sec. 4.3, we applied the $\mathcal{L}_{PDM}$ on the persistence diagrams of the original graph $G_0$ and the estimate $\hat{p}_{G_0}$ on the one-shot graph generation baselines to observe the general performance gap when utilizing $\mathcal{L}_{PDM}$. Shown in Table 3, we empirically demonstrate that $\mathcal{L}_{PDM}$ successfully guides baseline networks to produce topologically reliable graphs. Notably, the average metric values improved by $1.09 \sim 3.29$ times across all methods, highlighting the impact of $\mathcal{L}_{PDM}$.

Table 2: Ablation study on $\mu_{G_0}$. Gain refers to the performance gain obtained by adding $\mu_{G_0}$ to the denoising network with lower MMD discrepancy. $\lvert \bar{V} \rvert$ denotes the averaged number of nodes.

| Metric | ADNI $\lvert \bar{V} \rvert = 160, (\lvert V \rvert = 160)$ | | | ENZYMES $\lvert \bar{V} \rvert = 32.63, (10 \leq \lvert V \rvert \leq 125)$ | | | Community-small $\lvert \bar{V} \rvert = 15.28, (12 \leq \lvert V \rvert \leq 20)$ | | | Ego-small $\lvert \bar{V} \rvert = 6.41, (4 \leq \lvert V \rvert \leq 18)$ | | | Avg. Gain |
|---|---|---|---|---|---|---|---|---|---|---|---|---|---|
| | w/o $\mu_{G_0}$ | with $\mu_{G_0}$ | Gain | w/o $\mu_{G_0}$ | with $\mu_{G_0}$ | Gain | w/o $\mu_{G_0}$ | with $\mu_{G_0}$ | Gain | w/o $\mu_{G_0}$ | with $\mu_{G_0}$ | Gain | |
| Deg.↓ | 0.399 | 0.160 | -0.239 | 0.017 | 0.003 | -0.014 | 0.106 | 0.048 | -0.058 | 0.007 | 0.012 | 0.005 | **-0.096** |
| Clus.↓ | 1.006 | 0.886 | -0.120 | 0.049 | 0.045 | -0.004 | 0.123 | 0.068 | -0.055 | 0.039 | 0.024 | -0.015 | **-0.084** |
| Orbit↓ | 0.267 | 0.237 | -0.030 | 0.146 | 0.015 | -0.131 | 0.020 | 0.004 | -0.016 | 0.021 | 0.010 | -0.011 | **-0.061** |
| Avg.↓ | 0.557 | 0.427 | -0.130 | 0.071 | 0.021 | -0.050 | 0.083 | 0.040 | -0.043 | 0.022 | 0.015 | 0.007 | **-0.080** |

Table 3: Ablation study on $\mathcal{L}_{PDM}$. The superior values are highlighted in **bold**.

| Method | ADNI | | | | ENZYMES | | | | Community-small | | | | Ego-small | | | |
|---|---|---|---|---|---|---|---|---|---|---|---|---|---|---|---|---|
| | Deg.↓ | Clus.↓ | Orbit↓ | Avg.↓ | Deg.↓ | Clus.↓ | Orbit↓ | Avg.↓ | Deg.↓ | Clus.↓ | Orbit↓ | Avg.↓ | Deg.↓ | Clus.↓ | Orbit↓ | Avg.↓ |
| EDP-GNN | 1.063 | 1.430 | 0.626 | 1.039 | **0.052** | 0.895 | 0.474 | 0.474 | 0.056 | **0.038** | 0.069 | 0.054 | 0.029 | 0.046 | 0.008 | 0.028 |
| EDP-GNN+$\mathcal{L}_{PDM}$ | **1.011** | **0.854** | **0.600** | **0.822** | 0.134 | **0.729** | **0.143** | **0.335** | **0.041** | 0.041 | **0.015** | **0.032** | **0.024** | **0.041** | **0.007** | **0.024** |
| GDSS | 0.949 | 1.104 | **0.165** | 0.739 | 0.314 | **0.506** | 0.084 | 0.301 | 0.033 | 0.112 | 0.009 | 0.051 | 0.045 | 0.076 | **0.008** | 0.043 |
| GDSS+$\mathcal{L}_{PDM}$ | **0.403** | **0.675** | 0.539 | **0.539** | **0.127** | 0.529 | **0.058** | **0.238** | **0.026** | **0.096** | **0.005** | **0.043** | **0.040** | **0.059** | 0.010 | **0.036** |
| DiGress | 0.504 | 1.168 | 0.379 | 0.683 | 0.023 | 0.051 | 0.205 | 0.093 | 0.089 | 0.091 | 0.049 | 0.076 | 0.026 | 0.090 | 0.023 | 0.046 |
| DiGress+$\mathcal{L}_{PDM}$ | **0.399** | **1.006** | **0.267** | **0.557** | **0.017** | **0.049** | **0.146** | **0.071** | **0.059** | **0.089** | **0.045** | **0.064** | **0.007** | **0.039** | **0.021** | **0.022** |
| LocalPPGN | 0.777 | 0.149 | 0.954 | 0.627 | 0.037 | 0.068 | 0.048 | 0.051 | 0.034 | 0.218 | **0.018** | 0.090 | 0.014 | 0.091 | **0.006** | 0.037 |
| LocalPPGN+$\mathcal{L}_{PDM}$ | **0.374** | **0.120** | **0.695** | **0.396** | **0.025** | **0.061** | **0.031** | **0.039** | **0.030** | **0.154** | 0.019 | **0.068** | **0.012** | **0.081** | 0.009 | **0.034** |
| TAGG (w/o $\mathcal{L}_{PDM}$) | 0.379 | 1.263 | 0.247 | 0.630 | 0.010 | 0.049 | 0.148 | 0.069 | **0.047** | 0.079 | 0.028 | 0.051 | 0.005 | 0.060 | 0.017 | 0.027 |
| TAGG | **0.160** | **0.886** | **0.237** | **0.427** | **0.003** | **0.045** | **0.015** | **0.021** | 0.048 | **0.068** | **0.004** | **0.040** | 0.012 | **0.024** | **0.010** | **0.015** |

Regardless of the size of the graph, the ablation studies presented in Tab. 2 and Tab. 3 demonstrate that our proposed method, which utilize the topology-aware self-attention module and PDM loss, consistently improves performance across the overall dataset. However, in the case of small graphs, its limited number of simplices results in a restricted set of topological features, which may diminish the effectiveness of our method when compared to larger graph datasets. As a result, the MMD metric results (degree, clustering, orbit) in Tab. 1 may not show significant differences. Nevertheless, the averaged MMD scores outperform all baselines on both the community-small and ego-small datasets, highlighting the meaningful impact of our method on generation performance of small graphs.

Moreover, the improvements in clustering and orbit metrics emphasize the effectiveness of $\mu_{G_0}$ and $\mathcal{L}_{PDM}$ in preserving essential topological structures. The improved clustering metrics demonstrate TAGG's ability to capture local connectivity patterns, while the improved orbit metrics reflect its capacity to preserve structural patterns in each node's local subgraphs. These findings validate the significant contribution of the proposed topology-aware framework in generating high-fidelity graphs.

## 5.3 Qualitative analysis

**Visualization of generated graphs.** We qualitatively compare TAGG with other baselines via visualization of the generated samples. As seen in Fig. 3, TAGG produces more realistic graphs that closely resemble test graphs across various datasets. More visualization of the generated graphs of TAGG on benchmark datasets can be found in Appendix D. Baseline models [4, 23, 32, 43] often failed to capture the nuanced patterns present in benchmark datasets. Particularly, TAGG better

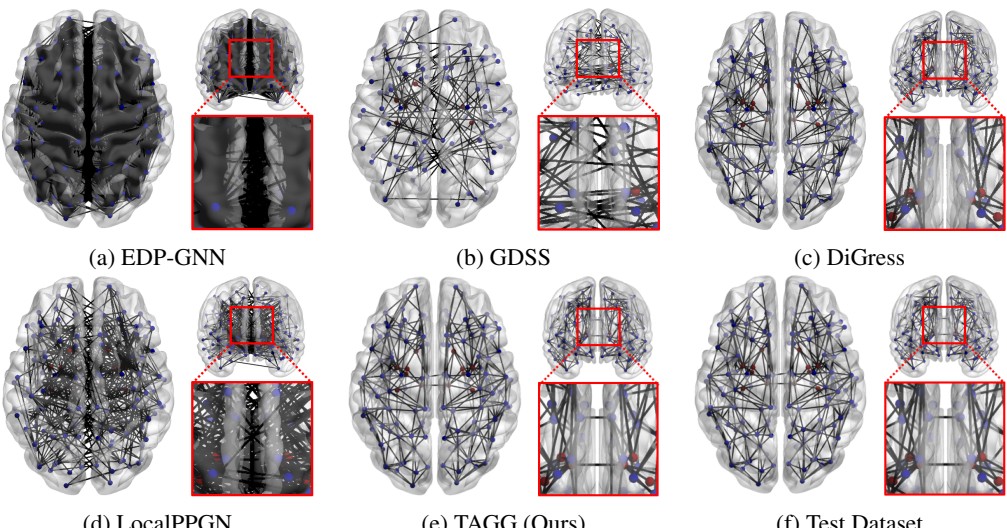

(a) EDP-GNN    (b) GDSS    (c) DiGress

(d) LocalPPGN    (e) TAGG (Ours)    (f) Test Dataset

Figure 3: Visualization of the averaged brain network of 50 samples from (a) EDP-GNN, (b) GDSS, (c) DiGress, (d) LocalPPGN, (e) TAGG, and (f) test dataset. The close-up box (red) highlights the inter-hemisphere connections, a key structural property of a brain network. The global structure, e.g., the sparsity and the inter-hemisphere connectivity, are well preserved using TAGG.

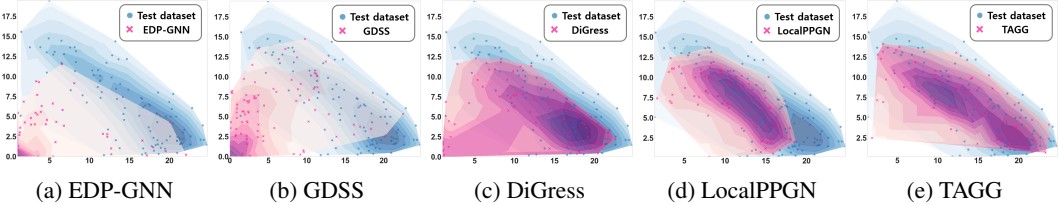

|     |     |     |     |     |
|:---:|:---:|:---:|:---:|:---:|
| (a) EDP-GNN | (b) GDSS | (c) DiGress | (d) LocalPPGN | (e) TAGG |

Figure 4: ATOL visualization of homological features on a 2D plane, where each point represent samples from test (Blue) and generated (Pink) ADNI dataset. Graphs generated with TAGG shows the highest resemblance.

represents critical characteristics compared to baselines, i.e., sparse inter-hemisphere connections and symmetry of hemispheres. Although the brain networks generated from the other discrete diffusion method, i.e., DiGress [43] , also exhibit high quality, they failed to capture the inter-hemisphere connection, a subtle but critical structural information of a brain network. Also in a topological perspective, the existence of inter-hemisphere connection determines *the number of connected components*, and TAGG is capable of capturing such topological structures. This demonstrates the effectiveness of topology-aware learning in capturing both global and detailed structural properties.

## 5.4 Homological assessment

To further validate the homological (dis)similarity between the test and the generated graphs, we utilize three distinct measures: 1) Automatic Topologically-Oriented Learning (ATOL) [34], 2) Persistence Image (PI) [2], and 3) Curvature Filtrations (CF) [41]. ATOL and PI are established vectorization methods in TDA that encode the homological features from the persistence diagrams into vectors of desired dimensions, and CF quantifies the (dis)similarity between graph distributions based on their topological structures. Below are the ADNI results. See Appendix F for more results.

**Quantification.** In the evaluation setting, we compute multivariate kernel density estimation with Kullback-Leibler (KL) divergence for ATOL, and Mean Squared Error (MSE) for PI, to quantify the distance between reference and generated graph distributions. Forman-Ricci Curvature was utilized in CF as the basis for curvature filtration to measure the topological differences.

Table 4: Quantified homological assessment.

|          | ATOL ↓ | PI ↓  | CF ↓    |
|----------|:------:|:-----:|:-------:|
| EDP-GNN  | 16.25  | 15.39 | 1115.00 |
| GDSS     | 1.43   | 3.18  | 154.62  |
| DiGress  | 0.33   | 0.67  | 31.68   |
| LocalPPGN| 0.60   | 0.91  | 45.75   |
| TAGG     | **0.23** | **0.59** | **27.96** |

As shown in Tab. 4, TAGG consistently exhibited the lowest dissimilarity values across all three metrics, indicating superior topological alignment with the reference graphs compared to all baseline methods. Specifically, compared to the second best method, i.e., DiGress, TAGG showed 30.3%, 11.9% and 11.7% reduction on ATOL KL-divergence, PI MSE, and CF, respectively.

Additionally, as discussed in Sec. 3, Betti numbers are the key descriptor of the topological characteristics of a graph. To quantitatively assess this aspect, we report the averaged Betti numbers (mean of 50 generated graphs) for each model, as shown in Table 5.

Given that the real brain network exhibits $\bar{\beta}_0 = 2.98$ and $\bar{\beta}_1 = 225.64$, TAGG closely approximates these real topological features, yielding $\bar{\beta}_0 = 2.92$ and $\bar{\beta}_1 = 229.96$, thereby demonstrating strong homological similarity. In contrast, most other models generally yield higher $\bar{\beta}_0$ values (around 5-6) and exhibit a wide variation in $\bar{\beta}_1$ values, deviating more significantly from the real data.

Table 5: Averaged Betti numbers of 0-, 1-dimension, i.e., $\bar{\beta}_0$, $\bar{\beta}_1$.

|          | $\bar{\beta}_0$ | $\bar{\beta}_1$ |
|----------|:-------:|:--------:|
| Test     | **2.98** | **225.64** |
| EDPGNN   | 1.00    | 1120.64  |
| GDSS     | 6.54    | 90.98    |
| DiGress  | 5.76    | 238.45   |
| LocalPPGN| 6.00    | 148.50   |
| TAGG     | **2.92** | **229.96** |

**Visualization.** We visualize the homological feature vectors of each sample obtained by ATOL on a 2-dimensional plane to evaluate the impact of our topology-aware learning. As shown in Fig. 4, the proximity between the blue (test) and pink (generated) points in (e) is far closer than in (a)-(d), indicating higher topological alignment and showcasing the impact of our topology-aware generation.

# 6 Conclusion

In this study, we proposed a novel graph generation framework preserving the intricate topology of the network. Through the proposed topology-aware attention module and the Persistence Diagram Matching Loss, we achieve high generation performance while maintaining the essential topological features of the original graphs. This approach improves the fidelity of generated graphs and provides valuable insights into their structure for both synthetic and real graph datasets. Our research addresses a critical challenge in complex real-world graph generation, particularly in the context of brain networks, and pave the way for practical graph generation with topological consistency.

**Limitation.** Despite the various methods on visually and empirically assessing the effect of topology-aware learning, e.g., ATOL, and ablation studies, it is hard to track the influence of individual topological features on the training of the denoising network. In addition, we utilized some of the most representative vectorization methods on persistence homology, i.e., persistence landscapes, which can be changed into other methods. Therefore, a more thorough investigation regarding other various vectorization methods may be done for future studies. (Partially discussed in Appendix C.)

## Acknowledgments and Disclosure of Funding

This research was supported by the National Research Foundation of Korea(NRF) grant funded by the Korea government(MSIT) (No. RS-2025-02216257, 50%; RS-2022-NR070486, 10%), Institute of Information & communications Technology Planning & Evaluation (RS-2022-II220290, 10%; IITP-2025-RS-2024-00437866, 10%), AI Graduate Program at POSTECH (RS-2019-II191906, 10%), and Samsung Electronics Co., Ltd (Project Code: IO240508-09825-01, 10%).

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

# A  Additional details

## A.1  On forward process

The additional details of the forward process described in Sec. 4.1, i.e., obtaining the noised graph $G_t$ from the original graph $G_0$, are provided. The notations below are reused from those introduced in Sec. 4. Note that we denoted the forward diffusion process at each time step, which imposes noise, as transition matrices $Q_t$, where $t = 1, 2, \ldots, T$. Each element of the matrices, $[Q_t]_{\eta, \xi}$, represents the probability that state $\eta$ changes to state $\xi$ as the time step changes from $t - 1$ to $t$, i.e., $[Q_t^V]_{\eta^V, \xi^V} = q(v^t = \xi^V \mid v^{t-1} = \eta^V)$ and $[Q_t^E]_{\eta^E, \xi^E} = q(e^t = \xi^E \mid e^{t-1} = \eta^E)$.

A noised graph $G_t = (V_t, E_t)$ from the previous graph $G_{t-1}$ can be obtained by sampling the type of nodes and edges from the categorical distribution after transition, which is derived as:

$$q\left(G_t \mid G_{t-1}\right) = \left(V_{t-1}Q_t^V, E_{t-1}Q_t^E\right). \tag{13}$$

More precisely, each row vector of $V_{t-1} \in \mathbb{R}^{N \times F_V}$ contains the node attributes of the graph $G_{t-1}$, as a form of one-hot row vector. By performing matrix multiplication $V_{t-1}Q_t^V$, we impose the noise to $V_{t-1}$ and obtain a new matrix, where each row vector represents the categorical distribution after the transition. Subsequently, we finally obtain the noised node attributes $V_t$ by sampling the type of nodes from the calculated categorical distribution $V_{t-1}Q_t^V$. The transition of the edge attributes is processed similarly to that of the node attributes.

The forward process from time $t - 1$ to time $t$ is mathematically formulated as follows:

$$
\begin{aligned}
q\left(G_t \mid G_{t-1}\right) &= \left(V_{t-1}Q_t^V, E_{t-1}Q_t^E\right) \\
&= \left(\begin{bmatrix} \mathrm{Cat}\left(v_t^1; v_{t-1}^1 Q_t^V\right) \\ \vdots \\ \mathrm{Cat}\left(v_t^N; v_{t-1}^N Q_t^V\right) \end{bmatrix}, \begin{bmatrix} \mathrm{Cat}\left(e_t^{1,1}; e_{t-1}^{1,1} Q_t^E\right) & \cdots & \mathrm{Cat}\left(e_t^{1,N}; e_{t-1}^{1,N} Q_t^E\right) \\ \vdots & \ddots & \vdots \\ \mathrm{Cat}\left(e_t^{N,1}; e_{t-1}^{N,1} Q_t^E\right) & \cdots & \mathrm{Cat}\left(e_t^{N,N}; e_{t-1}^{N,N} Q_t^E\right) \end{bmatrix}\right),
\end{aligned}
\tag{14}
$$

where $\mathrm{Cat}(z; \omega)$ denotes the categorical distribution over one-hot row vector $z$ with a probability vector $\omega$, and the dimensions of $\omega$ are $F_V$ and $F_E$ for node and edge, respectively.

Moreover, assuming the Markovian property of the process, we can derive the transition matrix from time 0 to time $t$ by simply multiplying the individual transition matrices: $\bar{Q}_t^V = Q_1^V \cdot Q_2^V \cdots Q_t^V$, and $\bar{Q}_t^E = Q_1^E \cdot Q_2^E \cdots Q_t^E$. Then, similar to Eq. (14), the noised graph $G_t$ can also be obtained from time 0 by sampling from the distribution $q\left(G_t \mid G_0\right)$, which is mathematically formulated as follows:

$$
\begin{aligned}
q\left(G_t \mid G_0\right) &= \left(V_0\bar{Q}_t^V, E_0\bar{Q}_t^E\right) \\
&= \left(\begin{bmatrix} \mathrm{Cat}\left(v_t^1; v_0^1 \bar{Q}_t^V\right) \\ \vdots \\ \mathrm{Cat}\left(v_t^N; v_0^N \bar{Q}_t^V\right) \end{bmatrix}, \begin{bmatrix} \mathrm{Cat}\left(e_t^{1,1}; e_0^{1,1} \bar{Q}_t^E\right) & \cdots & \mathrm{Cat}\left(e_t^{1,N}; e_0^{1,N} \bar{Q}_t^E\right) \\ \vdots & \ddots & \vdots \\ \mathrm{Cat}\left(e_t^{N,1}; e_0^{N,1} \bar{Q}_t^E\right) & \cdots & \mathrm{Cat}\left(e_t^{N,N}; e_0^{N,N} \bar{Q}_t^E\right) \end{bmatrix}\right).
\end{aligned}
\tag{15}
$$

## A.2 On training algorithm of TAGG

The overall scheme of TAGG is demonstrated in Algorithm 1. The homological feature vector $\mu_{G_0}$ is obtained via persistence landscape of the original graph $G_0$, which is derived from the filtration of $G_0$. Using the resultant $\mu_{G_0}$ as an additional input, the denoising network of TAGG iteratively utilize the topology-aware attention module (TAM) to update the node and edge embeddings. The homological feature $\mu_{G_0}$ enhance the attention module to estimate topology-aware node and edge embeddings, i.e., $V_t$ and $E_t$, which leads to high fidelity of generated graphs. Consequently, the denoising network outputs a probability vector $\hat{p}_{G_0} = (\hat{p}_{V_0}, \hat{p}_{E_0})$ of the original graph, which is then optimized using the Cross-Entropy and Persistence Diagram Matching loss.

---

**Algorithm 1** Overall scheme of TAGG

---

1: **Input:** Original graph $G_0 = (V_0, E_0)$, number of diffusion step $T$, number of TAM $M$, hyperparameter $\alpha_1$ and $\alpha_2$.

2: **1. Obtain homological feature** $\mu_{G_0}$
3: $\mathrm{ph}_{G_0} \leftarrow \mathrm{Filtration}(G_0)$
4: Obtain persistence barcodes and persistence diagram $\mathcal{D}_{G_0}$ from $\mathrm{ph}_{G_0}$
5: $\mu_{G_0} = \mathrm{PersistenceLandscape}(\mathcal{D}_{G_0})$       ▷ Homological feature of the given graph $G_0$.

6: **2. Training TAGG**
7: Sample $t \sim \mathcal{U}(1, 2, ..., T)$
8: Sample noisy graph $G_t = (V_t, E_t) \sim \left(V_0 \bar{Q}_t^V, E_0 \bar{Q}_t^E\right)$
9: **Model input:** $G_t = (V_t, E_t), \mu_{G_0}$

10: **2-1. Estimate $\hat{p}_{G_0}$ via TAGG**
11: Given $V_t^0 = V_t$ and $E_t^0 = E_t$,
12: **for** $m = 0$ **to** $M - 1$ **do**
13:      $\mathcal{Q}_t^m, \mathcal{K}_t^m, \mathcal{V}_t^m \leftarrow \mathrm{Lin}_q(V_t^m), \mathrm{Lin}_k(V_t^m), \mathrm{Lin}_v(V_t^m)$     ▷ $d_k$ is the dimension of $\mathcal{Q}_t$ and $\mathcal{K}_t$.
14:      $\mathbf{Z}_{V_t^m} = \left(\mathcal{Q}_t^m \otimes \mathcal{K}_t^m\right) / \sqrt{d_k}$                  ▷ Self-attention of node features.

15:      $\mathbf{Z}_{E_t^m}, \mathcal{H}_{G_0} \leftarrow \mathrm{Lin}_e(E_t^m), \mathrm{Lin}_\mu(\mu_{G_0})$
16:      $\mathcal{A}_t^m \leftarrow \mathbf{Z}_{V_t^m} \odot \mathbf{Z}_{E_t^m} + \mathcal{H}_{G_0}$       ▷ Utilize $\mu_{G_0}$ as a global attention bias term.
17:      $\mathcal{A}_t^m \leftarrow \mathrm{Softmax}(\mathcal{A}_t^m)$            ▷ Compute topology-aware attention score.

18:      $V_t^{m+1}, E_t^{m+1} \leftarrow \mathcal{N}_V(\mathcal{A}_t^m \odot \mathcal{V}_t^m), \mathcal{N}_E(\mathcal{A}_t^m)$
19: **end for**

20: $\hat{p}_{G_0} = (\hat{p}_{V_0}, \hat{p}_{E_0}) = \left(\mathrm{LayerNorm}(V_t^M), \mathrm{LayerNorm}(E_t^M)\right)$

21: **Model output:** probability of denoised graph $\hat{p}_{G_0}$

22: **2-2. Training Objective**
23: $\mathcal{L}_{\mathrm{final}} = \mathcal{L}_{\mathrm{CE}}^V(G_0, \hat{p}_{G_0}) + \alpha_1 \mathcal{L}_{\mathrm{CE}}^E(G_0, \hat{p}_{G_0}) + \alpha_2 \mathcal{L}_{PDM}(G_0, \hat{p}_{G_0})$

### A.3 On implementation

We provide additional details of experiment settings used in TAGG. As explained in Sec. 4.3, the training objective of TAGG has two real valued hyperparameters $\alpha_1 \in (0, 1]$ and $\alpha_2 \in (0, 1]$, each used to control the cross-entropy loss of edges $\mathcal{L}_{\mathrm{CE}}^E$ and the persistence diagram matching (PDM) loss $\mathcal{L}_{PDM}$, respectively. The hyperparameters $\alpha_1$ and $\alpha_2$ were chosen through a grid search of values in $\{1, 0.1, 0.01, 0.001, 0.0001\}$ on each dataset, and the settings are shown in Tab. 6. We followed the hyperparameters provided in the original papers for the baseline methods. For the datasets that were not included in the original papers, we conducted the same hyperparameter search as with TAGG to ensure a fair comparison. Additionally, after the reverse diffusion process to sample the generated graphs, we quantize the entries of the adjacency matrices using the operator $1_{x>0.5}$, resulting in a binary adjacency matrix.

Table 6: Hyperparameters of TAGG on different datasets

| Hyperparameter | ADNI | ENZYMES | Community-small | Ego-small |
|:---:|:---:|:---:|:---:|:---:|
| $\alpha_1$ | 1 | 1 | 0.001 | 0.01 |
| $\alpha_2$ | 0.001 | 0.0001 | 0.001 | 0.0001 |

### A.4 On additional ablation studies

We conducted an additional ablation study on PDM loss to provide a homological assessment to disentangle the effects of our core components as below. As mentioned in Sec. 5.4, each metric measures the topological difference, meaning that a lower score is better.

Table 7: Quantitative comparison of ATOL, PI, and CF metrics across models. Lower values indicate better performance.

| Method | ATOL | PI | CF |
|:---|:---:|:---:|:---:|
| EDP-GNN | 16.25 | 15.39 | 1115.00 |
| EDP-GNN+$\mathcal{L}_{\mathrm{PDM}}$ | **16.22** | **14.91** | **1038.69** |
| GDSS | 1.43 | 3.18 | 154.62 |
| GDSS+$\mathcal{L}_{\mathrm{PDM}}$ | **1.28** | **2.90** | **124.30** |
| DiGress | 0.33 | 0.67 | 31.68 |
| DiGress+$\mathcal{L}_{\mathrm{PDM}}$ | **0.27** | **0.63** | **28.80** |
| LocalPPGN | 0.60 | 0.91 | 45.75 |
| LocalPPGN+$\mathcal{L}_{\mathrm{PDM}}$ | **0.52** | **0.89** | **42.20** |

The result clearly shows that PDM loss ensures topological alignment, leading to improvement in all metrics, which supports the results of the manuscript.

## A.5 On additional experiments

To further demonstrate the robustness and superiority, we have conducted additional experiments on the standard Planar and SBM datasets.

Table 8: Performance comparison on Planar and SBM datasets. Lower is better.

| Method | Planar | | | | SBM | | | |
|---|---|---|---|---|---|---|---|---|
| | Deg. | Clus. | Orbit | Avg. | Deg. | Clus. | Orbit | Avg. |
| GraphRNN | 0.234 | 1.429 | 1.239 | 0.967 | 1.459 | 1.797 | 0.988 | 1.414 |
| GDSS | 0.232 | 1.032 | 1.137 | 0.800 | 0.991 | 1.705 | **0.484** | 1.060 |
| DiGress | 0.106 | 0.314 | 1.430 | 0.617 | 0.242 | 0.744 | 1.344 | 0.777 |
| TAGG | **0.058** | **0.311** | **1.056** | **0.475** | **0.112** | **0.476** | 0.798 | **0.462** |

As evident from the table, TAGG consistently achieves superior or highly comparable performance across all evaluated metrics on both the Planar and SBM datasets. This further supports and strengthens the experimental results presented in the main manuscript.

In addition, validity, novelty, and uniqueness are common metrics used in molecular graph generation, mainly to validate their molecular properties. The table below presents these additional metrics (results of SPECTRE are from its original paper [29]):

Table 9: Comparison of additional metrics on Planar and SBM datasets. Higher is better.

| Method | Planar | | | SBM | | |
|---|---|---|---|---|---|---|
| | Val. | Uniq. | Nov. | Val. | Uniq. | Nov. |
| GDSS | 33.3 | 100 | 100 | 33.7 | 100 | 99.0 |
| DiGress | 38.8 | 100 | 93.6 | 23.1 | 100 | 100 |
| SPECTRE | 47.5* | 100* | 100* | 60.0* | 100* | 100* |
| TAGG | **62.5** | 100 | 100 | **63.6** | 100 | 100 |

Additionally, to further demonstrate our model's capabilities, we evaluated the diversity of TAGG-generated ADNI graphs using averaged Graph Edit Distance (GED) and Average Pairwise Distance (APD). GED quantifies structural diversity by measuring the number of operations (node/edge additions or deletions) required to transform one graph into another [1]. APD measures pairwise distances based on graph properties, such as degree distributions, where higher values indicate greater diversity. The results are as follows: TAGG-generated graphs show (GED: 1.55 / APD: 0.23), while test graphs show (GED: 1.12 / APD: 0.15). The increased GED and APD of TAGG-generated graphs ($\sim 40\%$) compared to test graphs suggests that the generated graphs exhibit sufficient diversity while remaining within a valid distribution.

# B  Analysis of computational and practical feasibility

We provide a detailed feasibility analysis including the graph generation process and persistent homology component. We report the measured time (sec/epoch) and memory usage (in MB) for TAGG and baseline diffusion models, i.e., ConGress (continuous) and DiGress (discrete) [43] on four datasets of varying scale. All experiments were conducted on a single GeForce RTX 3090 with 24GB of GPU memory, with batch size 4.

Table 10: Averaged sec/epoch of 10 epochs.

| sec/epoch | ADNI | ENZYMES | Community-small | Ego-small |
|---|---|---|---|---|
| ConGress [43] | 11.2 | 9.4 | 1.4 | 2.7 |
| DiGress [43] | 7.7 | 7.6 | 1.1 | 1.9 |
| TAGG | 77.1 | 65.0 | 2.4 | 3.0 |

The primary sources of computational burden in TAGG can be decomposed into three aspects: **1)** the computation of the homological feature $\mu$, **2)** persistent homology, and **3)** the Wasserstein distance in the PDM loss. While $\mu$ can be pre-computed, requiring minimum additional computation during training, the computational cost of persistent homology and the Wasserstein distance is comparatively high, accounting for most of the time differences shown in the table above. This limitation is primarily due to the CPU-based implementation of both methods, a significant challenge faced by researchers in the TDA community, as none of the existing Python libraries currently support GPU computation. Despite the well-known high computational complexity of persistent homology, we believe this is still one of the highly attractive research topics to investigate in the graph machine learning community.

In the case for large graph datasets (e.g., ENZYMES with up to 125 nodes, and ADNI with 160 nodes), the number of simplices increases leading to the additional costs for computing the persistent homology, e.g., the graph filtration. Despite this, the training times on ENZYMES and ADNI dataset are 64.8 and 78.1 seconds per epoch, respectively, and fully training the TAGG (1000 epochs) can be done within a day (< 22h) on both datasets.

Table 11: Averaged inference time (in sec) for generating 4 samples on 5 individual runs

| sec | ADNI | ENZYMES | Community-small | Ego-small |
|---|---|---|---|---|
| DiGress [43] | 63.18 | 14.48 | 14.27 | 13.92 |
| TAGG | 64.19 | 15.04 | 14.44 | 14.99 |

To verify the practicality of our model, we also report the average inference time (sec) for generating 4 graph samples over 5 independent runs across all datasets, using a single GPU. As shown in Tab. 11, the inference time of TAGG is comparable to that of the baseline model, i.e., Digress, across all datasets. This demonstrates that TAGG achieves superior performance and fidelity in graph generation, as reported in Sec.5, without incurring any significant additional computational cost.

In addition, as shown in Tab. 12, the memory requirements for TAGG are not substantial. Considering that these results were achieved using a single GPU, we think that the computational demand is practically feasible for real-world applications especially when multiple GPUs are employed.

Table 12: Required memory for training (MB), batch size 4

| MB | ADNI | ENZYMES | Community-small | Ego-small |
|---|---|---|---|---|
| DiGress [43] | 5680 | 5988 | 1570 | 1532 |
| TAGG | 6186 | 6352 | 1580 | 1542 |

# C Topological design choices

In TAGG, we utilized node degree as the 0-simplex filter function as in Eq. 2 and persistent landscape as the vectorization method of persistence diagrams. However, we do not claim in any way that such design choice is the optimal or state-of-the-art framework for topology-aware graph generation. While the choices can vary, the change in generation performance is minor, which we will cover in the following subsections.

## C.1 Choice of filter functions

Below, we report the generation performance using different filtration functions: degree, clustering coefficient, and betweenness centrality.

Table 13: Comparison of different filter functions.

| Function | ADNI | | | | Community-small | | | | Ego-small | | | |
|---|---|---|---|---|---|---|---|---|---|---|---|---|
| | Deg. | Clus. | Orb. | Avg. | Deg. | Clus. | Orb. | Avg. | Deg. | Clus. | Orb. | Avg. |
| degree | 0.160 | 0.886 | 0.237 | 0.427 | 0.048 | 0.068 | 0.004 | 0.040 | 0.012 | 0.024 | 0.010 | 0.015 |
| clust coef | 0.242 | 0.802 | 0.198 | 0.414 | 0.061 | 0.057 | 0.008 | 0.042 | 0.016 | 0.030 | 0.015 | 0.020 |
| btw_cent | 0.252 | 0.847 | 0.236 | 0.445 | 0.051 | 0.059 | 0.017 | 0.042 | 0.015 | 0.036 | 0.010 | 0.020 |

As can be seen, the choice of filtration function does not have a crucial impact on the overall performance.

## C.2 Choice of vectorization methods

We also evaluated both persistence landscapes (PL) and persistence images (PI) on Community-small and Ego-small datasets.

Table 14: Comparison of different vectorization methods.

| | Community-small | | | | Ego-small | | | |
|---|---|---|---|---|---|---|---|---|
| | Deg. | Clus. | Orb. | Avg. | Deg. | Clus. | Orb. | Avg. |
| PL | 0.048 | 0.068 | 0.004 | 0.040 | 0.012 | 0.024 | 0.010 | 0.015 |
| PI | 0.050 | 0.079 | 0.004 | 0.044 | 0.011 | 0.025 | 0.016 | 0.017 |

Results indicate minimal performance differences, suggesting our method is robust to different vectorization choices.

# D Additional visualization

In this section, we provide additional visualizations that have not been included in the main paper due to space limit.

## D.1 Generated graph samples via GraphRNN

For example, Fig. 5 includes the (a) brain network and (b) ENZYMES graphs generated via GraphRNN [47], a baseline method of our paper. Align with the qualitative comparison results shown in Sec. 5 of the main paper, TAGG better represents critical characteristics compared to GraphRNN.

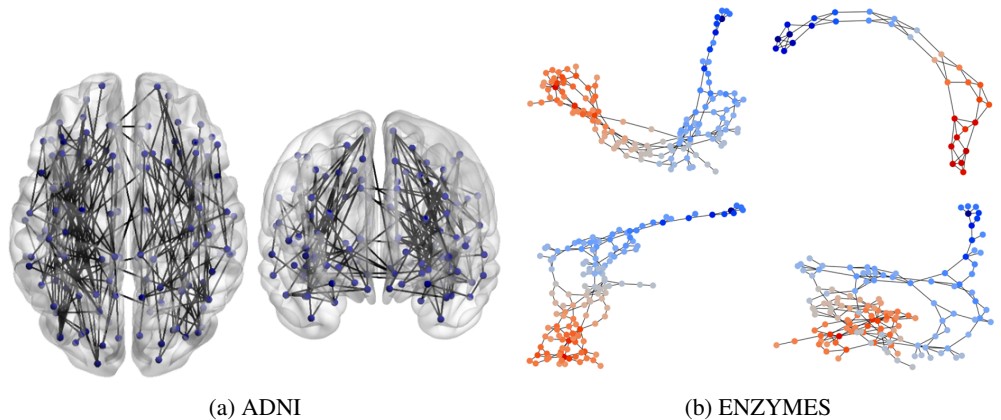

(a) ADNI                                    (b) ENZYMES

Figure 5: Visualization of the generated graphs using GraphRNN; (a) Brain network from ADNI dataset and (b) ENZYMES graphs.

## D.2 Generated graph samples via TAGG

Also, Fig. 6,7,8 shows additional graphs generated using TAGG to validate the consistency of the generation performance.

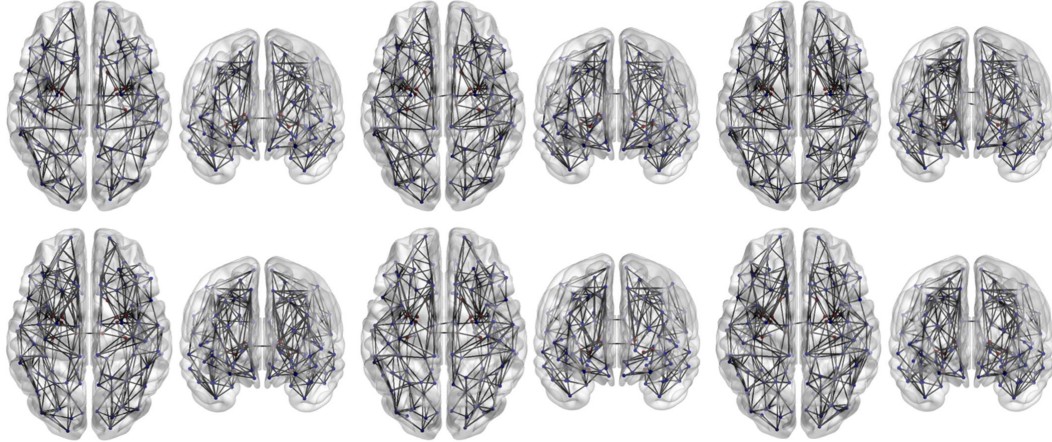

Figure 6: Visualization of the brain network generated using TAGG. TAGG successfully generates homologically reliable brain network, preserving the symmetry of brain network and the edges interconnecting left and right hemisphere.

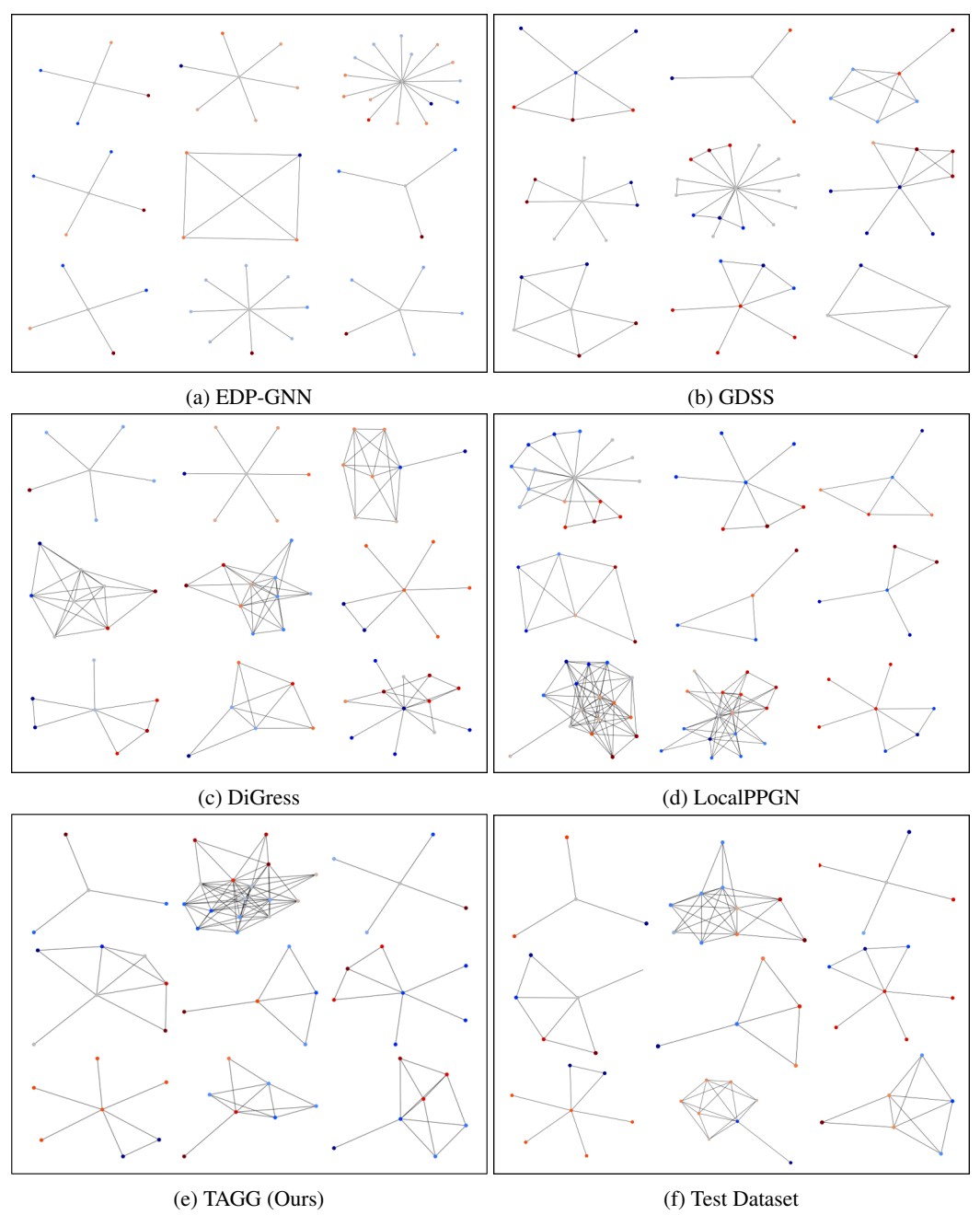

(a) EDP-GNN

(b) GDSS

(c) DiGress

(d) LocalPPGN

(e) TAGG (Ours)

(f) Test Dataset

Figure 7: Visualization of the generated Ego-small graphs from (a) EDP-GNN, (b) GDSS, (c) DiGress, (d) LocalPPGN, (e) TAGG, and (f) test dataset.

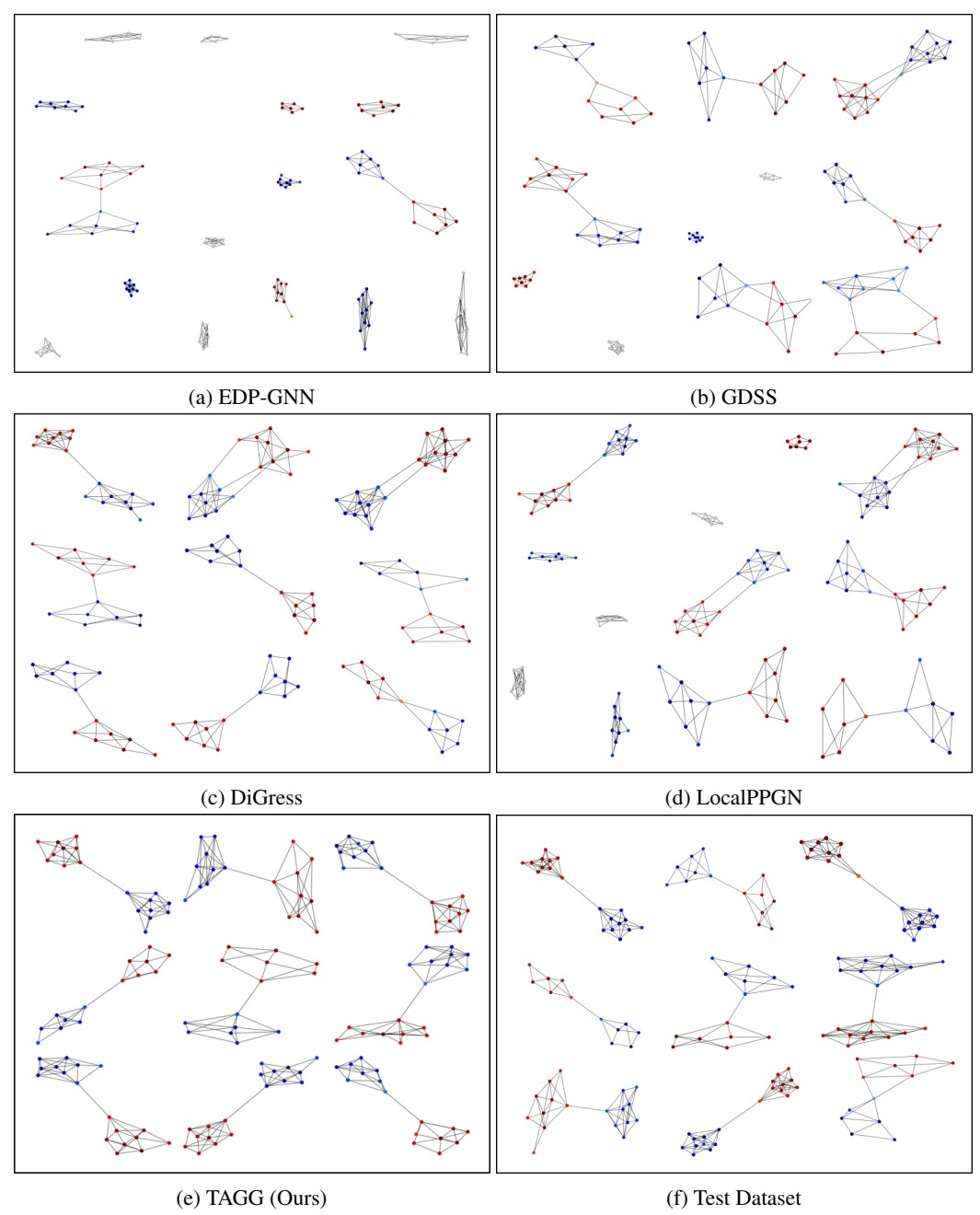

Figure 8: Visualization of the generated Community-small graphs from (a) EDP-GNN, (b) GDSS, (c) DiGress, (d) LocalPPGN, (e) TAGG, and (f) test dataset. Compared to the baselines, TAGG generates the most topologically equivalent graphs. TAGG successfully generates an edge that connects two communities.

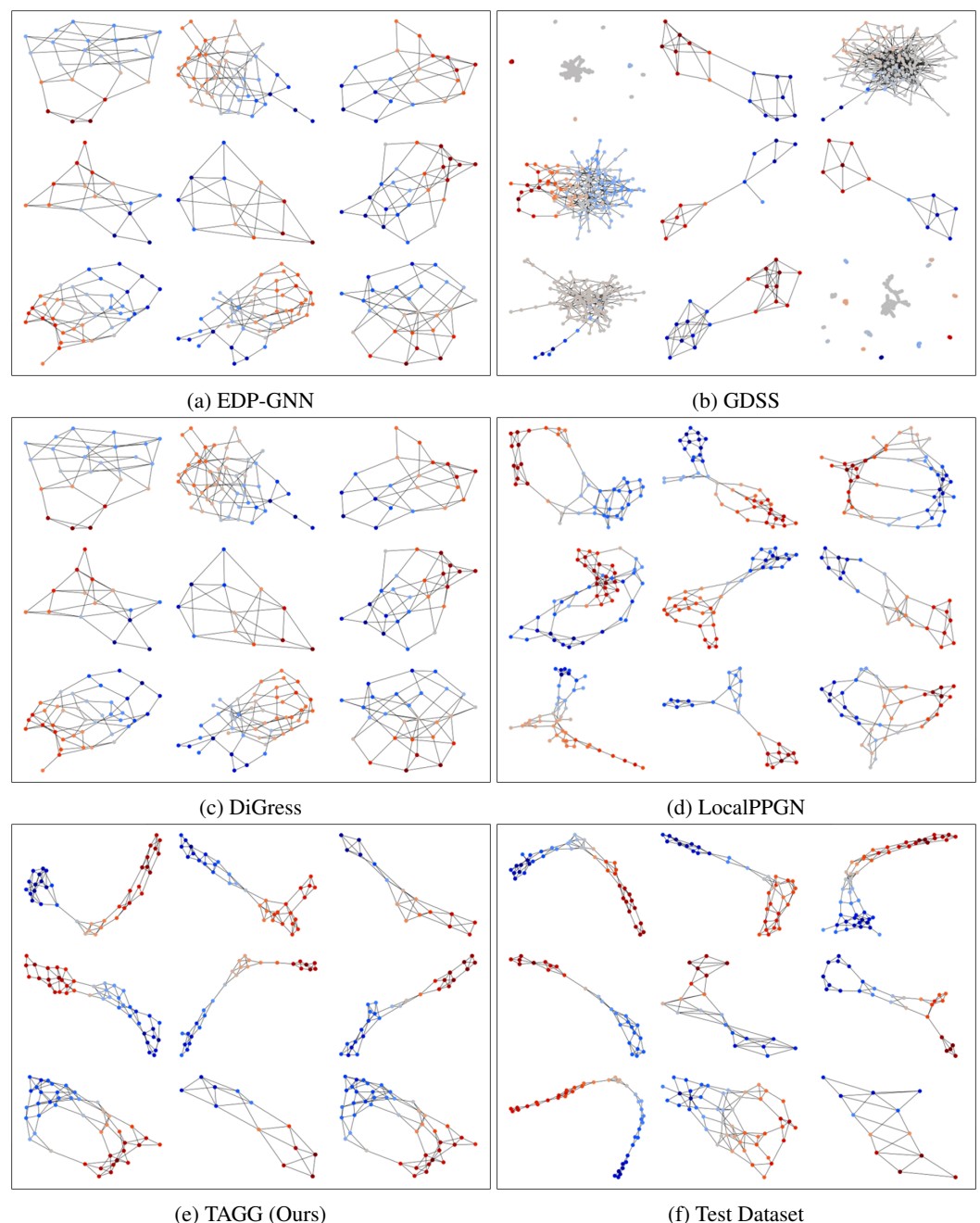

Figure 9: Visualization of the generated ENZYMES graphs from (a) EDP-GNN, (b) GDSS, (c) DiGress, (d) LocalPPGN, (e) TAGG, and (f) test dataset. TAGG can generate topologically equivalent graphs.

## D.3 ATOL result of all baseline methods and TAGG

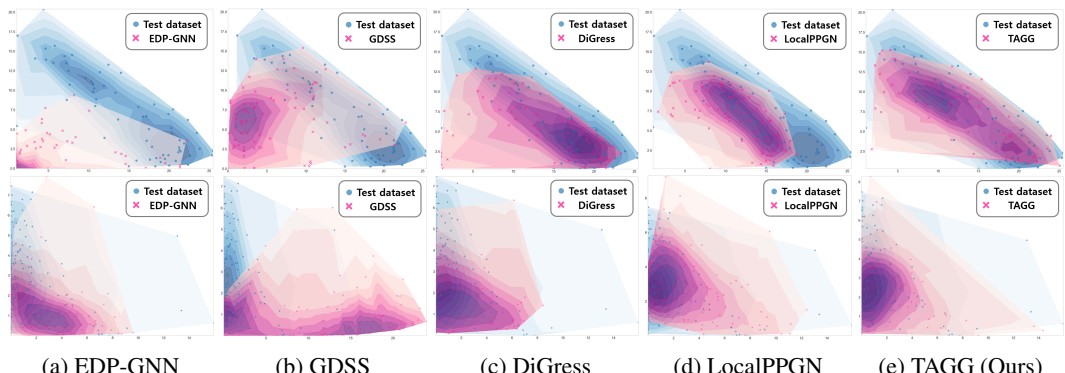

(a) EDP-GNN     (b) GDSS     (c) DiGress     (d) LocalPPGN     (e) TAGG (Ours)

Figure 10: ATOL visualization of homological features derived from the test (Pink) and generated graphs (Blue). Top: ADNI dataset, Bottom: ENZYMES dataset. The distribution of features from a sample from TAGG exhibit the best similarity with the ground truth.

# E    Visualization of hidden representations

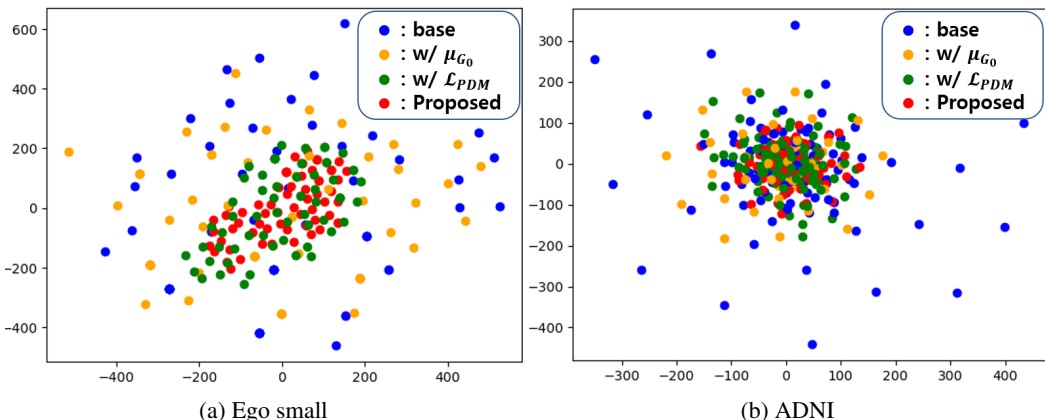

(a) Ego small                                    (b) ADNI

Figure 11: t-SNE visualization of the trained features from the topology-aware attention module on (a) Ego-small and (b) ADNI dataset. Colors denote the t-SNE results from trained features under different model settings; Blue: baseline, Orange: TAGG without $\mu_{G_0}$, Green: TAGG without $\mathcal{L}_{PDM}$, and Red: TAGG.

In order to investigate the effect of the topology-aware learning framework, Fig. 11 demonstrates the t-SNE visualization of the trained features from the topology-aware attention module on different datasets. Specifically, the hidden features from the final layer of the attention module were extracted and projected onto a 2-dimensional plane using t-SNE, providing a visual representation of the trained hidden features in the latent space. To evaluate the individual contributions of the topology-aware attention module and the PDM loss of TAGG, t-SNE results were obtained using the same models from the ablation study, along with a baseline model that excludes both $\mu_{G_0}$ and $\mathcal{L}_{PDM}$. Notably, incorporating either the homological feature $\mu_{G_0}$, the PDM loss, or both consistently improved performance across all datasets. In line with the enhanced quantitative results, the visualization reveals differences in the latent space between the cases where neither method was applied and where both were utilized, indicating that the graph features were optimized into a more desirable latent space.

# F    Additional quantified homological assessment

Table 15: Quantified homological assessment.

| Methods | ADNI | | | ENZYMES | | |
|---|---|---|---|---|---|---|
| | ATOL (KL-div)↓ | PI (MSE)↓ | CF ↓ | ATOL (KL-div)↓ | PI (MSE)↓ | CF ↓ |
| EDP-GNN | 16.25 | 15.39 | 1115.00 | 2.98 | 21.35 | 87.66 |
| GDSS | 1.43 | 3.18 | 154.62 | 1.12 | 0.13 | 72.76 |
| DiGress | 0.33 | 0.67 | 31.68 | 0.55 | 0.06 | 32.76 |
| LocalPPGN | 0.60 | 0.91 | 45.75 | 0.61 | 0.04 | 55.74 |
| TAGG | **0.23** | **0.59** | **27.96** | **0.29** | **0.01** | **21.29** |

To further validate the homological assessment of TAGG, we provide additional quantified result on the ADNI and ENZYMES datasets. As shown in Tab. 15, TAGG consistently exhibited the lowest dissimilarity values across all three metrics on both datasets, showcasing superior topological alignment with the reference graphs compared to all baseline methods. Specifically, on the ADNI dataset, TAGG showed about 30.3%, 11.9% and 11.7% reduction on ATOL KL-divergence, PI MSE, and CF, respectively, compared to the second best scores. Similarly, on the ENZYMES dataset, TAGG yielded improvements of around 52.7%, 25.0%, and 64.9% on the same metrics, demonstrating consistent topological performance.

