# OpenReview forum: "Topology-aware Graph Diffusion Model with Persistent Homology"
_NeurIPS.cc/2025/Conference — NeurIPS 2025 poster_

### Official Review · Reviewer_zh2S · 2025-06-30

**Clarity:** 3
**Significance:** 3
**Originality:** 3
**Rating:** 4
**Confidence:** 4

**Summary:**

This manuscript introduces a method to integrate persistent homology into graph diffusion models. The approach comprises three steps: (1) compute the persistence landscape from input graphs; (2) embed the landscape into the diffusion backbone via an attention module; and (3) augment the base diffusion loss (as in [38]) with an additional term comparing persistence diagrams.

**Questions:**

1. The loss in Equation (11) involves an optimization subproblem. How do you compute gradients through this inner optimization during backpropagation?
2. In Line 103, you describe “adding the node $v_1$,” but multiple nodes may be added simultaneously. Could you clarify this step in the algorithm?
3. In Section 5.4, TAGG’s superior homology preservation is expected by design. Can you perform an ablation study that adds only the persistence-loss term to other baseline models to disentangle the contributions of topology loss and attention fusion?

**Ethical Concerns:**

["NO or VERY MINOR ethics concerns only"]

**Final Justification:**

I think this work is solid after the authors addressed my and other reviewers' concerns by providing more ablative studies. These help with clarifying what is effective in the proposed method.

**Limitations:**

Besides the limitations pointed out by the authors, I'd like to bring up one another possible limitation: the computation of persistence diagrams seem to limited to graphs with small sizes and maybe this is worth pointing out in the paper.

**Quality:**

3

**Strengths And Weaknesses:**

The overall idea of incorporating global information of a graph into the generative model is nice. The proposed method of using the persistence landscape might be a good way to do so. The presentation is clear and easy to follow and the problem is important in the context of graph generative models.

However, the idea of using persistence diagrams in graph models is not new. For example, this work [1] already explored the idea of using persistence diagrams in message passing neural networks. I understand that the authors are using a different approach by combining the persistence landscape with an attention module, but there is no justification of why this is better than using persistence image + message passing as done in [1]. I think at least some empirical comparison is needed to justify the proposed method.

[1] Zhao et al. 2020, Persistence Enhanced Graph Neural Network, AISTATS

---

> ### Author Rebuttal · Authors · 2025-07-31
>
> We sincerely appreciate your detailed review. We will address the reviewer's concerns for further improvements.
>
> [1] Zhao et al., "Persistence Enhanced Graph Neural Network." AISTATS, 2020
>
> [2] Maria, Clément, et al. "The gudhi library: Simplicial complexes and persistent homology." Mathematical Software–ICMS, 2014
>
> ---
> **W1) Why is TAGG better than [1]?**
>
> **A)** A) We appreciate the reviewer’s reference to [1] and agree that persistent homology has been previously explored in graph learning. However, our work departs significantly from prior approaches such as [1], which focused on graph classification and incorporated persistence images as input features into message passing neural networks. In contrast, our method targets graph generation and leverages topological signals at a deeper architectural level; by introducing TAM that dynamically modulates the denoising process, and by enforcing topological alignment via PDM loss. These components are embedded directly into the generative model, rather than simple feature-level augmentation, offering a principled and structure-aware approach to topology-driven generation.
> We also agree that empirical comparison of vectorization methods is valuable. We therefore evaluated both persistence landscapes (PL) and persistence images (PI) on Community-small and Ego-small datasets (larger graphs will be reported in the Appendix):
>
> |   |Com-s||||Ego-s||||
> |---|---|---|---|---|---|---|---|---|
> |   |Deg.|Clus.|Orb.|Avg.|Deg.|Clus.|Orb.|Avg.|
> |PL|0.048|0.068|0.004|0.040|0.012|0.024|0.010|0.015|
> |PI|0.050|0.079|0.004|0.044|0.011|0.025|0.016|0.017|
>
> Results indicate minimal performance differences, suggesting our method is robust to different vectorization choices. While we do not claim PL or any specific filter function to be universally superior, our novelty lies in recasting graph generation from a topological perspective and effectively employing TDA tools (TAM and PDM) to ensure topology-aware generation, rather than in the selection of a single vectorization strategy.
>
> ---
> **Q1) How do you compute gradients through this inner optimization (Eq. 11) during backpropagation?**
>
> **A)** Addressing the concern about the backpropagation, the 1-Wasserstein distance used in Equation (11) is implemented based on a differentiable optimal transport method by leveraging Gudhi library [2]. Also, the homological feature vector is fed to the MLP layer before incorporating it to the node and edge features in the attention module, which ensures the end-to-end training.
>
> ---
> **Q2) In Line 103, you describe “adding the node v_1,” but multiple nodes may be added simultaneously. Could you clarify this step in the algorithm?**
>
> **A)** Thank you for the great insight and question. As you mentioned, multiple nodes with the same filter value can be added simultaneously in the filtration process. In this case, following the sublevel set filtration, their incident edges are also included in the evolving subgraph at that filtration step. While Line 103 offers a simplified exposition, the underlying process correctly handles such additions. Notably, this does not limit the homological information; persistence barcodes are inherently recorded as a multiset of intervals, which perfectly captures all births, deaths, and persistence of homological features occurring at any given filtration value, regardless of simultaneous additions. For the readers' better understanding, we will add additional explanation in the Appendix.
>
> ---
> **Q3) Can you perform an ablation study that adds only the persistence-loss term to other baseline models?**
>
> **A)** Thank you for pointing out an important analysis. As you correctly pointed out, disentangling the effects of topology preservation and attention design is essential for a thorough understanding of TAGG’s performance.
> To address this, we conducted the exact ablation study you suggested, i.e., adding only the persistence-loss term to existing baseline models, and present the results in Table 3. As shown, incorporating the PDM loss alone leads to consistent improvements across not only baselines but also all datasets, demonstrating the effectiveness of PDM loss, independent of our attention design.
>
> ---
> **L1) Persistence diagram computation may not scale to large graphs.**
>
> **A)** We appreciate your insightful suggestion regarding this potential limitation. We agree that the computational complexity of persistent homology generally increases with graph size. However, as discussed in Appendix B, the datasets used in our work span a range of graph sizes (average node counts from 6.41 to 160), and the computational burden for generating persistence diagrams proved manageable. Our experiments show that a complete TAGG training (1000 epochs) is feasible within a single day. Thus, we demonstrate that incorporating persistent homology is practical for real-world applications across various typical graph scales, even if not maximally efficient. We agree that this aspect warrants explicit mention and will include it in the limitations section of the manuscript.

---

> > ### Comment · Reviewer_zh2S · 2025-08-03
> >
> > I agree with the authors that specific vectorization method may not matter so much and some discussion along this line should be included in the paper.
> >
> > For Q3, I was asking whether an ablative study can be carried out specifically for the Quantified homological assessment. The ablative study in table 3 is not about the homological assessment. Such an ablative study on homological assessment could help disentangle the effects of the topological loss and the attention module.

---

> > > ### Author Response · Authors · 2025-08-06
> > >
> > > We appreciate your response. We hope the explanation below resolves your concerns.
> > >
> > > ---
> > > >**Some discussion along this line should be included in the paper.**
> > >
> > > **A)** Thank you for the considerate feedback. We agree that such discussion will help prevent misunderstanding of the core contribution of our work and provide a more complete picture to the readers. As suggested, we will add a discussion in the revised manuscript to clarify that our contribution is not in proposing the most suitable filter function or vectorization method, but in our novel use of graph topology in the generative process. We will also include experimental results in the appendix to show that there are only minor performance variations with different choices of filter functions and vectorization methods.
> > >
> > > ---
> > > >**Ablation study for the quantified homological assessment.**
> > >
> > > **A)** We apologize for the misunderstanding. As requested, we conducted an additional ablation study on PDM loss to provide a homological assessment to  disentangle the effects of our core components as below. As mentioned in Line 316-322, each metric measures the topological difference, meaning that a lower score is better.
> > >
> > > |   |ATOL|PI|CF|
> > > |---|---|---|---|
> > > |EDP-GNN|16.25|15.39|1115.00|
> > > |EDP-GNN$+\mathcal{L}_{PDM}$|**16.22**|**14.91**|**1038.69**|
> > > |GDSS|1.43|3.18|154.62|
> > > |GDSS$+\mathcal{L}_{PDM}$|**1.28**|**2.90**|**124.30**|
> > > |DiGress|0.33|0.67|31.68|
> > > |DiGress$+\mathcal{L}_{PDM}$|**0.27**|**0.63**|**28.80**|
> > > |LocalPPGN|0.60|0.91|45.75|
> > > |LocalPPGN$+\mathcal{L}_{PDM}$|**0.52**|**0.89**|**42.20**|
> > >
> > > The result clearly shows that PDM loss ensures topological alignment, leading to improvement in all metrics, which supports the results of the manuscript. We will add this result to the Appendix of the final manuscript.

---

> > > > ### Comment · Reviewer_zh2S · 2025-08-06
> > > >
> > > > Thank you for addressing my concerns and I will update my score accordingly.

---

### Official Review · Reviewer_hBMW · 2025-07-02

**Clarity:** 2
**Significance:** 2
**Originality:** 2
**Rating:** 4
**Confidence:** 4

**Summary:**

The paper presents TAGG, a topology-aware graph generation framework that integrates persistent homology into diffusion models to better account for topological/structural information of undirected graphs. In particular, the authors introduce a novel loss function ( persistence diagram matching PDM), which enforces topological similarity between generated and original graphs, and a topology-aware attention mechanism that augments the self-attention module with homological features. The proposed approach is evaluated across four datasets (two synthetic ones), including brain networks.

**Questions:**

1. Regarding homological similarity, could you report the distributions of betti_0 and betti_1 for the generated graphs from different models and test data?
2. How does the choice of vectorization and filtration function impact the overall performance? The method should work for any choice of vectorization, no?! What is the motivation for persistence landscapes?
3. Does the incorporation of $\mu_{G_0}$ in the attention layer lead to a constant additive term in Eq. (9) across all diffusion steps?
4. Could the authors report results with random $\mu_{G_0}$? I would like to know the sensitivity to that choice.
5. How does the proposed model compare with Digress on their evaluation setup?
6. In line 112, what if the homological feature does not disappear?

**Ethical Concerns:**

["NO or VERY MINOR ethics concerns only"]

**Final Justification:**

The proposed method demonstrates empirical improvements over DiGress and exhibits potential for advancing the current art in graph generation. I would like to see more experiments on molecular data, including additional evaluation metrics. Also, reporting results using different filtration functions and vectorization schemes might be useful to understand the sensitivity of the model to those choices.

**Limitations:**

The paper briefly discusses limitations in Section 6.

**Paper Formatting Concerns:**

No formatting issues.

**Quality:**

2

**Strengths And Weaknesses:**

Strengths

- The paper is well-written and clear;
- The application on brain networks is very interesting with many nice visualizations;
- The authors provide ablation studies to validate the effectiveness of their loss function and topology-aware attention mechanism.

Weaknesses

**Novelty**. I consider the novelty limited as:

- The core generation process closely follows the method in [1];
- Topology-aware loss functions based on PH have been abundant in graph representation learning [e.g., 2,3];
- TAM consists of a minor modification of a standard self-attention module, incorporating topological features.

[1] Digress: Discrete denoising diffusion for graph generation. ICLR, 2023.

[2] Improving Self-supervised Molecular Representation Learning using Persistent Homology. NeurIPS, 2023.

[3] Boosting Graph Pooling with Persistent Homology. NeurIPS, 2024.

**Experiments**. I have concerns regarding the completeness and significance of the experiments. In particular:
- The paper says it "performed three replicate experiments to report averaged performance". Could you report the variances?
- Important evaluation metrics are missing, such as novelty & uniqueness or spectrum (see [4] for others).
- Also, the paper lacks baselines for one-shot graph generation, such as SPECTRE [4].
- Overall, this paper would benefit from considering more datasets. I would like to see more results on molecular graphs, and assess how the proposal compares with SOTA for molecule generation (e.g., see [1]).

[4] SPECTRE: Spectral Conditioning Helps to Overcome the Expressivity Limits of One-shot Graph Generators. ICML, 2022.

**Motivation**. Overall, modeling graph distributions requires capturing a diverse range of structural patterns. I am not entirely convinced that better accounting for homological features (btw, I believe the paper refers to persistent (detailed) homological features) would lead to better models in general, considering the diversity of structural patterns. This is more pronounced because the topological information in the proposal is derived from a fixed filtration and a specific choice of vectorization. These design choices are not adequately motivated. In addition, only the average topological embedding (from training) is used --- it is unclear to me why such a simple modeling choice would work across different graph distributions.

---

> ### Author Rebuttal · Authors · 2025-07-31
>
> We sincerely appreciate your careful review. In the following, we will address all the reviewer's remaining doubts.
>
> [1] Digress: Discrete denoising diffusion for graph generation. ICLR, 2023.
>
> [2] SPECTRE: Spectral Conditioning Helps to Overcome the Expressivity Limits of One-shot Graph Generators. ICML, 2022.
>
> [3] Graph filtration learning. ICML, 2020.
>
> ---
> **W-N1) The core generation process follows [1].**
>
> **A)** Our novelty does not lie in introducing discrete diffusion but in proposing a topological perspective for graph generation. While we adopt a discrete diffusion backbone as [1], TAGG explicitly incorporates TDA tools via PDM loss and TAM to ensure topological alignment between generated and real graphs. This crucial capability, which preserves properties like connected components and loops, fundamentally distinguishes TAGG from prior works.
>
> ---
> **W-N2) Topology-aware loss functions have been abundant in graph representation learning.**
>
> **A)** We acknowledge that PH-based loss functions have been previously utilized. However, we claim that our PDM loss contributes a novel approach to graph generation by focusing on preserving global homological information during the generative process. Unlike previous methods primarily optimizing node-edge distributions, PDM ensures topological resemblance, capturing intrinsic shape and connectivity patterns crucial for high fidelity graph generation. Experiments validate that generated graphs retain key topological properties grounded in TDA.
>
> ---
> **W-N3) TAM consists of a minor modification of a standard self-attention module.**
>
> **A)** We humbly disagree that a modification of a standard self-attention module necessarily diminishes the novelty of our work. What information is integrated into attention, how it is computed, and its downstream impact are critical to model design. We emphasize that graph topology, specifically in terms of persistent homological features of various dimensions, has been largely unaddressed in the context of graph generation models. By carefully integrating these topological features into the attention mechanism, our TAM enables the module to explicitly account for global graph structure. This allows for the computation of attention scores that directly contribute to superior generation performance, as rigorously demonstrated in our experiments and ablation study.
>
> ---
> **W-E1) Variance of performance.**
>
> **A)** As requested, we provide the standard deviation for the performance metrics of our proposed TAGG model in the table below.
>
> ||Deg.|Clus.|Orbit||Deg.|Clus.|Orbit|
> |---|---|---|---|---|---|---|---|
> |ADNI|0.160±0.021|0.886±0.012|0.237±0.039|ENZYMES|0.003±0.001|0.045±0.002|0.015±0.004|
> |Com-s|0.048±0.021|0.068±0.006|0.040±0.009|Ego-s|0.012±0.006|0.024±0.011|0.010±0.005|
>
> As these results indicate, TAGG demonstrates consistently stable performance across different datasets and metrics. Due to character limits, we'll include the standard deviations for baseline methods in the Appendix of the final paper.
>
> ---
> **W-E2) Important evaluation metrics are missing.**
>
> **A)** Please refer to W4 of Reviewer Z9nJ.
>
> ---
> **W-E3) Lack baselines for one-shot graph generation, such as [2].**
>
> **A)** To ensure a fair comparison, our current set of baselines includes four well-known one-shot graph generation methods. Following your suggestion, we attempted to run [2] using its public code. Unfortunately, the substantial computational hurdles (as noted on its public GitHub) hindered us from reporting results within the deadline. However, DiGress [1] has already demonstrated superior results over [2] on both general and molecular graph benchmarks (SBM, Planar, Community-small, and QM9). Since TAGG consistently showed superior or comparable performance to DiGress (Table 1; W1, W4 of Reviewer Z9nJ), we can reasonably infer TAGG would also outperform [2].
>
> ---
> **W-E4) More datasets like molecular graphs will be beneficial.**
>
> **A)** To address the general capabilities of our method across diverse graph types, we conducted additional experiments on standard datasets (SBM and Planar, as detailed in W1 of Reviewer Z9nJ) and on a molecular graph dataset (QM9). Below, we report the results for QM9 dataset:
>
> |QM9|Val.|Uniq.|Nov.|
> |---|---|---|---|
> |ConGress|98.9|97|39.8|
> |DiGress|99.5|100|33.7|
> |TAGG|99.0|100|50.6|
>
> TAGG demonstrates superior or comparable performance across all metrics, notably achieving significantly higher novelty (+16.9%p) compared to DiGress. One possible explanation for this superior novelty is that, unlike standard discrete diffusion models that often overfit to frequent local motifs, our PDM loss encourages the exploration of diverse yet valid topologies. This helps the model avoid mode collapse and generate structurally novel graphs that still preserve key global properties, leading to the observed performance gains.
>
> ---
> **W-M1) Better accounting for homological features may not account for diverse structural patterns.**
>
> **A)** We acknowledge that our fixed filtration and vectorization choices might not capture *all* graph structural diversity. However, graph topology is a critical structural aspect largely overlooked in prior graph generation works, and persistent homology offers a principled way to encode these global structures.
> As noted in Sec. 6 (Limitations), our specific choices are widely adopted and well-established within the TDA community. Our results demonstrate that even with this straightforward design, our approach leads to meaningful and consistent improvements in both quantitative metrics (Tables 1–4) and qualitative analyses (Figures 3–4). Furthermore, our ablation studies confirm that each PH-based component significantly contributes to the observed performance gains.
>
> ---
> **W-M2) It is unclear whether a simple averaging of $\mu$ would work across different graph distributions.**
>
> **A)** Using the average $\mu$ is a deliberate choice that provides a representative topological summary of the dataset's general distribution. This simple yet effective strategy works by capturing the dominant topological patterns, guiding our model to produce graphs aligning with the dataset's characteristics. Our strong experimental performance across diverse datasets empirically validates its sufficiency.
> Crucially, this average $\mu$ encodes meaningful, learned topological information, which is fundamentally different from sampling from random noise (as detailed in Q4).
>
> ---
> **Q1) Distributions of betti_0 and betti_1.**
>
> **A)** We agree that the distribution of Betti numbers is crucial for understanding homological similarity, thus we report the averaged Betti numbers (mean of 50 graphs) below:
>
> |ADNI|Test|EDPGNN|GDSS|DiGress|LocalPPGN|TAGG|
> |---|---|---|---|---|---|---|
> |$\bar\beta_0$|**2.98**|1|6.54|5.76|6|**2.92**|
> |$\bar\beta_1$|**225.64**|1120.64|90.98|238.45|148.5|**229.96**|
>
> Given that the real brain network exhibits $\bar\beta_0=2.98$, our proposed TAGG model closely approximates these real topological features, yielding $\bar\beta_0=2.92$, demonstrating strong homological similarity. In contrast, most other models generally yield higher $\bar\beta_0$ (around 5-6), deviating from the real data.
>
> ---
> **Q2) Impact of the choice of vectorization and filtration function.**
>
> **A)** Below, we report the generation performance using different filtration functions: degree, clustering coefficient, and betweenness centrality. As can be seen, the choice of filtration function does not have a crucial impact on the overall performance, as well as the vectorization method (refer to W1 of Reviewer zh2S).
>
> ||ADNI|||ENZY.|||Com-s|||Ego-s|||
> |---|---|---|---|---|---|---|---|---|---|---|---|---|
> |   |Deg|Clus|Orb|Deg|Clus|Orb|Deg|Clus|Orb|Deg|Clus|Orb|
> |degree|0.160|0.886|0.237|0.003|0.045|0.015|0.048|0.068|0.004|0.012|0.024|0.010|
> |clust coef|0.242|0.802|0.198|-|-|-|0.061|0.057|0.008|0.016|0.030|0.015|
> |btw_cent|0.252|0.847|0.236|0.009|0.047|0.012|0.051|0.059|0.017|0.015|0.036|0.010|
>
> ---
> **Q3) Does the incorporation of in the attention layer lead to a constant additive term in Eq. (9) across all diffusion steps?**
>
> **A)** Yes, the homological embedding is injected to the attention computation as a global bias term, which is employed across all diffusion steps.
>
> ---
> **Q4) Random $\mu_{G_0}$ sensitivity.**
>
> **A)** We report results using random $\mu_{G_0}​$. Performance degrades notably across all datasets and metrics, which aligns with our design intuition: the averaged $\mu_{G_0}​$ serves as a proxy for general topological characteristics of the original distribution and guides the network to generate topologically faithful samples. Random $\mu_{G_0}​$ lacks such semantic structure, leading to poor topological alignment. Degradation is more severe for larger graphs (ADNI), likely due to the increased importance of embedded topological information.
>
> ||Com-s|||Ego-s|||ADNI|||
> |---|---|---|---|---|---|---|---|---|---|
> ||Deg|Clu|Orb|Deg|Clu|Orb|Deg|Clu|Orb|
> |avg|**0.048**|**0.068**|**0.004**|**0.012**|**0.024**|**0.010**|**0.160**|**0.886**|**0.237**|
> |rand|0.091|0.069|0.026|0.024|0.030|0.015|1.126|1.880|0.693|
>
> ---
> **Q5) How does the proposed model compare with Digress on their evaluation setup?**
>
> **A)** All comparisons strictly follow the evaluation setups suggested in the manuscript or using the public code, including DiGress. Additional hyperparameter details for TAGG are provided in Appendix A.3.
>
> ---
> **Q6) In line 112, what if the homological feature does not disappear?**
>
> **A)** When a homological feature does not disappear during the filtration, it is referred to as an essential feature, represented as barcodes of the form $(b, \infty)$. In the context of graphs, essential 0- and 1-dimensional features typically correspond to connected components and loops, respectively.
> In practice, since $\infty$ cannot be directly represented in computation, it is approximated using a sufficiently large constant or the maximum filtration value.

---

> > ### Comment · Reviewer_hBMW · 2025-08-06
> >
> > Thank you for your detailed rebuttal. Overall, I still have concerns regarding the novelty of this work. However, the additional results, ablations, and clarifications provided have alleviated most of my initial concerns. Therefore, I will update my score.

---

### Official Review · Reviewer_7g2a · 2025-07-02

**Clarity:** 4
**Significance:** 3
**Originality:** 3
**Rating:** 5
**Confidence:** 4

**Summary:**

Classical methods for operations on graphs, such as random graph generation or graph denoising, sometimes scramble important graph features that we need. One of these features is the topology of the graph. The paper does a good job addressing this problem.

**Questions:**

Can these methods be generalized to general simplicial complexes rather than just graphs?

**Ethical Concerns:**

["NO or VERY MINOR ethics concerns only"]

**Limitations:**

yes

**Paper Formatting Concerns:**

I see no formatting issue

**Quality:**

3

**Strengths And Weaknesses:**

The paper is well written and seems to make an important contribution.

---

> ### Author Rebuttal · Authors · 2025-07-31
>
> **Q1) Can these methods be generalized to general simplicial complexes?**
>
> **A)** We thank the reviewer for this insightful question. While our current framework is designed for graphs, viewed as 1-dimensional simplicial complexes, the underlying methodology can, in principle, be extended to more general simplicial complexes. For example, in settings such as 3D point cloud data, one could construct higher-dimensional simplicial complexes and compute higher-order persistent homology accordingly. Although a direct application is currently not feasible due to the input constraints of the network architecture, the core idea of discrete diffusion-based generation guided by topological alignment remains applicable in broader settings with suitable architectural adaptations.

---

### Official Review · Reviewer_Z9nJ · 2025-07-05

**Clarity:** 2
**Significance:** 3
**Originality:** 3
**Rating:** 5
**Confidence:** 3

**Summary:**

The authors introduce a graph generation method based on discrete diffusion, defined as topology-aware because it also preserves the structural features of the original graphs. The approach uses a loss function based on Persistence Diagram Matching (PDM) and integrates a Topology-aware Attention Module within the denoising network.
Several experiments are presented, with a focus on the application to complex brain network data, showing the method’s effectiveness in real-world scenarios.

**Questions:**

Please see above

**Ethical Concerns:**

["NO or VERY MINOR ethics concerns only"]

**Final Justification:**

After considering the submission and my initial review, I confirm my accept recommendation.
The approach is well-motivated and the experiments, particularly on complex brain networks, demonstrate its effectiveness in real-world scenarios.

**Limitations:**

Missing runtime analysis

**Paper Formatting Concerns:**

-

**Quality:**

3

**Strengths And Weaknesses:**

*Strengths*
- The issue of reconstructing the homological features of the original graphs is very important, and the proposed method is therefore interesting because it explicitly addresses this aspect. The experiments on homological assessment are also very interesting.

*Weaknesses*

- Persistent homology works better and makes more sense on graphs that are not small. The experiments on Ego-Small are therefore particularly challenging and appreciated. However, I would have included, at least in the appendix, experiments on standard datasets such as SBM and Planar. I expect the model to achieve performance at least on par with other methods on these datasets.
- The computational cost analysis is done in comparison with DiGress, which is known to be a discrete diffusion method and relatively slow. To have a more informative comparison, I suggest including at least one continuous diffusion method. This point should also be stressed in the limitations.
- Persistent homology is always a difficult topic, and since it is allowed, I would add in the appendix a richer explanation in terms of graphical support for Section 3 (Preliminaries), to help readers who are less familiar with topology.
- The experiments are missing standard metrics such as Validity, Novelty, and Uniqueness, which are common in graph generation works. Even if they are not directly related to topology, they are useful to provide a more complete picture of the generative model’s capabilities.
- In the sentence
>“Recently, diffusion-based methods [3, 20, 38] showed promising capability in graph generation, by defining the forward and reverse diffusion processes and training a neural network that mimics the reverse process to reconstruct the graphs.”

I suggest adding two recent references that propose methods based on diffusion in the graph spectrum, which also capture both local and global structural information:

- Martinkus, K., Loukas, A., Perraudin, N., & Wattenhofer, R. (2022, June). Spectre: Spectral conditioning helps to overcome the expressivity limits of one-shot graph generators. In ICML.
- Minello, G., Bicciato, A., Rossi, L., Torsello, A., & Cosmo, L. (2025). Generating graphs via spectral diffusion. In ICLR.

---

> ### Author Rebuttal · Authors · 2025-07-31
>
> We sincerely thank the reviewer Z9nJ for the constructive comments. In the following, we will address all the reviewer's remaining doubts.
>
> [1] SPECTRE: Spectral Conditioning Helps to Overcome the Expressivity Limits of One-shot Graph Generators. ICML, 2022.
> [2] An exact graph edit distance algorithm for solving pattern recognition problems. ICPRAM, 2015.
> [3] Generating graphs via spectral diffusion. ICLR, 2025.
>
> ---
> **W1) Additional experiments on standard datasets, as PH works better on large graphs.**
>
> **A)** Great insight on the effect of PH, as mentioned in Line 285-287. Our initial choice of datasets was driven by the goal of demonstrating TAGG’s scalability across a diverse range of graph sizes (average nodes ranging from 6.41 to 160). To further demonstrate its robustness and superiority as requested, we have conducted additional experiments on the standard SBM and Planar datasets.
>
> |    |   Planar   ||||   SBM   ||||
> |----|------------|---|---|----|------------|---|---|----|
> |    |Deg.|Clus.|Orbit|Avg.|Deg.|Clus.|Orbit|Avg.|
> |GraphRNN|0.234|1.429|1.239|0.967|1.459|1.797|0.988|1.414|
> |GDSS|0.232|1.032|1.137|0.800|0.991|1.705|**0.484**|1.060|
> |DiGress|0.106|0.314|1.430|0.617|0.242|0.744|1.344|0.777|
> |TAGG|**0.058**|**0.311**|1.056|**0.475**|**0.112**|**0.476**|0.798|**0.462**|
>
> As evident from the table, TAGG consistently achieves superior or highly comparable performance across all evaluated metrics on both the Planar and SBM datasets. This further supports and strengthens the experimental results presented in the main manuscript. We will include these results in the Appendix of the revised version.
>
> ---
> **W2) To have a more informative comparison, include at least one continuous diffusion method.**
>
> **A)** We thank the reviewer for this valuable suggestion regarding our computational cost analysis. We clarify that the purpose of the cost analysis was not to emphasize TAGG's superior efficiency, but rather to demonstrate its practical feasibility and applicability even with the computation of topological features. As requested, we have now included ConGress, a continuous diffusion method, to the comparison table below (results are averaged sec/epoch of 10 epochs):
>
> | |comm-small|ego-small|ENZYMES|ADNI|
> |---|---|---|---|---|
> |ConGress|1.4|2.7|9.4|11.2|
> |DiGress|1.1|1.9|7.6|7.7|
> |TAGG|2.4|3|65|77.1|
>
> It is important to note that discrete diffusion models, like DiGress, generally require fewer computation steps compared to continuous diffusion models, which typically involve discretizing numerous small steps to simulate the continuous process. This fundamental distinction often results in discrete methods having faster training and inference speeds, as our expanded comparison illustrates. We will elaborate on these points in the limitations section and Appendix B of the revised version.
>
> ---
> **W3) Adding a richer explanation, i.e., graphical support, of PH in the appendix will help readers’ understandings.**
>
> **A)** We fully agree that adding more explanations will improve the reader’s understanding of the contribution of our work and its effect on graph generation. Frankly, the explanation of PH is never enough, which is why we referred to the solid paper on TDA in Line 84. However, the reviewer’s  suggestion to add graphical support is perhaps a very good idea, and we will add figures demonstrating the filtration process and the computation of persistent diagrams in the Appendix.
>
> ---
> **W4) Missing standard metrics such as Validity, Novelty, and Uniqueness.**
>
> **A)** We thank the reviewer for pointing out these additional standard metrics. We are aware that Validity, Novelty, and Uniqueness are common in molecular graph generation, especially for validating their molecular properties. Following the reviewer’s suggestion and acknowledging that some works (e.g., [1]) also report these metrics on synthetic datasets, we have now included results for Validity, Novelty, and Uniqueness on the SBM and Planar datasets. The table below presents these additional metrics (results of SPECTRE are from its original paper):
>
> |   |Planar|||SBM|||
> |---|---|---|---|---|---|---|
> |   |Val|Uniq|Nov|Val|Uniq|Nov|
> |GDSS|33.3|100|100|33.7|100|99.0|
> |DiGress|38.8|100|93.55|23.1|100|100|
> |SPECTRE|47.5*|100*|100*|60*|100*|100*|
> |TAGG|62.5|100|100|63.6|100|100|
>
> Additionally, to further demonstrate our model's capabilities, we evaluated the diversity of TAGG-generated ADNI graphs using averaged Graph Edit Distance (GED) and Average Pairwise Distance (APD). GED quantifies structural diversity by measuring the number of operations (node/edge additions or deletions) required to transform one graph into another [2]. APD measures pairwise distances based on graph properties, such as degree distributions, where higher values indicate greater diversity. The results are as follows: TAGG-generated graphs show (GED: 1.55 / APD: 0.23), while test graphs show (GED: 1.12 / APD: 0.15). The increased GED and APD of TAGG-generated graphs (~40%) compared to test graphs suggests that the generated graphs exhibit sufficient diversity while remaining within a valid distribution.
>
> ---
> **W5) Suggest adding two recent references that propose methods based on diffusion in the graph spectrum.**
>
> **A)** We thank the reviewer for this valuable suggestion. We agree that incorporating these two recent references regarding diffusion methods in the graph spectrum [1,3] will enrich the discussion in our related work section (Line 103), providing readers with a more comprehensive overview of graph diffusion models. We will include these references in the revised manuscript.
>
> ---
> **L1) Missing runtime analysis**
>
> **A)** Please refer to the W2) and Appendix B.

---

> > ### Comment · Reviewer_Z9nJ · 2025-08-06
> >
> > I thank the authors for addressing the concerns raised and for providing additional experimental results.

---

### Decision · Program_Chairs · 2025-09-17

**Decision:**

Accept (poster)

**Comment:**

The paper was reviewed by four expert reviewers.
The reviewers generally agree that the paper is well-written and mostly appreciate the novelty of incorporating topological features into a diffusion graph generative model. Compelling experimental results are supported by both real and synthetic datasets, with quantitative and qualitative evaluations demonstrating the performance of TAGG in comparison to multiple baselines.
The authors provided a rebuttal that addressed the concerns raised in the reviews. After the rebuttal and disucssion, the reviewers were satisfied with the justifications and details provided in the rebuttal and recommended acceptance.

The reviews provided valuable feedback to the authors, which the authors should carefully consider to improve the final version of their paper, in addition to experiments, additional evaluation measures, run-time, and ablations conducted during the discussion period, which should be added to the final version of the paper.